# Intracellular tension sensor reveals mechanical anisotropy of the actin cytoskeleton

Sorosh Amiri [1,2], Camelia Muresan[1,3], Xingbo Shang[1,3], Clotilde Huet-Calderwood[4], Martin A. Schwartz [3,5,6], David A. Calderwood [4,5] & Michael Murrell [1,3,7] ✉

The filamentous actin (F-actin) cytoskeleton is a composite material consisting of cortical actin and bundled F-actin stress fibers, which together mediate the mechanical behaviors of the cell, from cell division to cell migration. However, as mechanical forces are typically measured upon transmission to the extracellular matrix, the internal distribution of forces within the cytoskeleton is unknown. Likewise, how distinct F-actin architectures contribute to the generation and transmission of mechanical forces is unclear. Therefore, we have developed a molecular tension sensor that embeds into the F-actin cytoskeleton. Using this sensor, we measure tension within stress fibers and cortical actin, as the cell is subject to uniaxial stretch. We find that the mechanical response, as measured by FRET, depends on the direction of applied stretch relative to the cell's axis of alignment. When the cell is aligned parallel to the direction of the stretch, stress fibers and cortical actin both accumulate tension. By contrast, when aligned perpendicular to the direction of stretch, stress fibers relax tension while the cortex accumulates tension, indicating mechanical anisotropy within the cytoskeleton. We further show that myosin inhibition regulates this anisotropy. Thus, the mechanical anisotropy of the cell and the coordination between distinct F-actin architectures vary and depend upon applied load.

Mechanical forces are generated and transmitted throughout the F-actin cytoskeleton to drive essential biological functions such as cell division[1,2] and migration[3,4]. The F-actin cytoskeleton also senses and responds to external mechanical cues[5–7], which impacts diverse phenotypic states from differentiation[8] to the determination of cell fate[9]. Thus, understanding how the F-actin cytoskeleton mediates the generation and transmission of mechanical forces can further our understanding of homeostatic processes[10], development[11,12], and disease progression[13].

The F-actin cytoskeleton is a composite material, composed of cortical actin network and bundled F-actin stress fibers (SFs)[14–16]. Cortical F-actin is nucleated by Arp2/3 and formins[17], and is disordered in F-actin polarity, length, and orientation, which forms a thin, cross-linked network beneath the cell membrane[14,18–20]. Through the activity of myosin II molecular motors, it generates forces that drive cortical flows and shape changes[21–23]. By contrast, F-actin within SFs, nucleated by formin[24], is arranged into aligned, bipolar arrays, initially

[1]Systems Biology Institute, 850 West Campus Drive, Yale University, West Haven, CT 06516, USA. [2]Department of Mechanical Engineering and Material Science, 17 Hillhouse Ave, Yale University, New Haven, CT 06511, USA. [3]Department of Biomedical Engineering, 17 Hillhouse Ave, Yale University, New Haven, CT 06511, USA. [4]Department of Pharmacology, 333 Cedar St, Yale University, New Haven, CT 06510, USA. [5]Department of Cell Biology, 333 Cedar St, Yale University, New Haven, CT 06510, USA. [6]Yale Cardiovascular Research Center, 300 George St, New Haven, CT 06511, USA. [7]Department of Physics, 217 Prospect Street, Yale University, New Haven, CT 06511, USA. ✉e-mail: Michael.murrell@yale.edu

characterized by their cable-like organization[25, 26]. Like cortical F-actin, SFs are contractile through myosin motor activity, and transmit tractions to the extracellular matrix (ECM)[5, 7, 18, 27, 28]. Despite their apparent differences in organization, recent work has demonstrated that the two are not separate entities, but integrated structures that direct the generation and transmission of tractions to the ECM[14, 16, 29]. However, how disparate F-actin architectures coordinate to bear mechanical loads within the cell cytoskeleton is unclear.

To date, the measurement of cell-generated mechanical forces[30–33] is principally limited to those exerted against the ECM, within focal assemblies[29, 34], receptors, or receptor-bound linkers[29, 34–36]. These external traction forces reflect the integration of intracellular molecular forces within the cytoskeleton, transmitted to integrin-based adhesions[37]. From traction, internal forces can only be inferred, for example through force balance[38]. However, force balance requires assumptions of the mechanical properties of the cell and a constitutive relationship that relates stresses to strains. To this end, the cell is often assumed to be a continuum of homogenous and isotropic material, where stress is linearly related to strain[39–42]. However, mechanical anisotropies and nonlinearities are thought to play major roles in cellular mechanobiology, including in the strain-dependent gene expression in stem cells[43–46], regulation of YAP mechanotransduction[47], and GEF binding[48–50] that activates Rho GTPases to control cytoskeletal dynamics. Further, large-scale mechanical behaviors such as cortical flows are attributed to the anisotropic behavior of subcellular[51] and cellular mechanics[52]. Thus, assessing the extent of mechanical anisotropy in the cell, and the role of F-actin architectures in its establishment, requires measurement of mechanical forces within the cell cytoskeleton.

Here we present a Förster Resonance Energy Transfer (FRET)-based tension sensor, which enables high-precision measurement of molecular tension in space and time within diverse architectures of the F-actin cytoskeleton. First, we correlate molecular tensions measured by FRET with traction as measured by Traction Force Microscopy (TFM). In doing so, we explore the role of cell shape and applied mechanical strain in mediating mechanical anisotropy. Second, we segment the FRET signal that appears in SFs versus cortical actin, to understand the role of distinct cytoskeletal architectures in establishing mechanical anisotropy. In combination, these results demonstrate that while structurally integrated, the cooperation between distinct F-actin architectures in bearing and generating mechanical loads is dynamic, depends upon the external state of applied load, and determines the mechanical anisotropy of the cytoskeleton.

## Results

### The tension sensor localizes to diverse F-actin architectures

To infer the molecular tension within the actin cytoskeleton, we designed a genetically encoded FRET tension sensor (TS) based on Filamin-A, an actin-binding protein that crosslinks actin filaments within the cortex and stress fibers[29, 53, 54]. The TS utilizes the actin-binding domain (ABD) from Filamin-A, the FRET construct with F40 nano-spring between Ig15 and Ig16, followed by Ig24 (Fig. 1a). The sensor forms a dimer in which each monomer contains the EGFP (GFP, donor) – TagRFP (RFP, acceptor) fluorophores connected by the F40 domain (Fig. 1a and Supplementary Fig. 1 and "Methods" section). The F40 domain consists of an entropic nano-spring that forms a random coil in the absence of tension on each monomer, where the fluorophores are close together, and the emission of GFP (donor) excites RFP (acceptor) with a high efficiency (high FRET). Under tensile loads of up to 6 pN[22], the coil extends, and the fluorophores move apart, decreasing the ability of the donor to excite the acceptor, thereby decreasing the FRET efficiency (low FRET). By contrast, under compressive loads, the fluorophores may move closer together, increasing the FRET efficiency[55]. Thus, within each monomer, the tension sensor converts molecular forces to 'intramonomeric' FRET (Supplementary

Fig. 16). Furthermore, the molecular forces exerted on the dimeric tension sensor can lead to changes in 'intradimeric' FRET, observed between the pairs of monomers within each dimeric tension sensor (Supplementary Fig. 16). When embedded in F-actin assemblies, the TS measures molecular tensions generated and resisted by the cell cytoskeleton. For example, it can measure myosin-generated tension between nearby actin filaments to which it binds and crosslinks. Further, as a control, we designed a control tension sensor (CTS) construct, which lacks the dimerization domain and therefore does not report tension (Fig. 1b, d).

The sensors are expressed in U2OS cells (Methods). We recognize that expression of the TS may alter the actin network and its connectivity, which can manifest as changes in F-actin architectures via bundling, consequently impacting the generation of mechanical force in the cytoskeleton. To address this, we measure the changes in the F-actin architecture, and changes in traction as a function of sensor expression. To measure F-actin organization, we measure F-actin alignment order ($\langle \varphi \rangle$, Methods). To evaluate force generation, we measure the associated tractions by TFM ("Methods" section and Supplementary Fig. 9). We note that in TS expressing cells ('TS cells'), neither significantly changes as a function of sensor expression. Thus, we suggest expression of the tension sensor does not significantly perturb the cytoskeleton compared to non-expressing cells. However, CTS expressing cells ('CTS cells') show a higher diffuse (cytoplasmic) amount of sensor, compared to TS cells. We suggest this is due to the lack of dimerization domain in the CTS structure which reduces its ability to localize to F-actin in comparison to the TS (also shown in non-patterned cells, Supplementary Fig. 2).

First, we quantify the extent of spatial localization of the TS sensor to distinct F-actin architectures. To do so, we adhere U2OS cells expressing the sensors on polyacrylamide gels (E=2.8 kPa) with fibronectin micropatterns of different shapes (Methods)[4, 56, 57]. As has been reported previously[58], cells patterned in circular shapes have a radial distribution of SFs (Fig. 1c, d). By contrast, cells patterned into squares and triangles assemble SFs on the sides of the cell (Supplementary Fig. 3). Focal adhesion (paxillin) sites were concentrated at the periphery of all patterns. However, focal adhesions were also present within the center of the cell on circular shapes but not square or triangular shapes (Fig. 1c, d and Supplementary Fig. 3). To quantify the localization of the sensors to the actin, we conducted a Pearson's colocalization analysis ("Methods" section) between the immuno-fluorescence staining of actin using phalloidin and the RFP of the sensors across the whole cell (Fig. 1i). Both constructs showed a strong (for TS ≈0.84±.037 and for CTS ≈0.75 ± .09) and significant ($p$-value < 0.0001) correlation between the sensor and actin, indicating a higher colocalization of TS to the actin compared to the localization of the CTS to the actin. As we use confocal microscopy which captures a thin ~500 nm optical z-slice, we expect to image predominantly cortical actin, stress fibers, and intracellular networks near the basal surface, but not the center of the cell. Thus, henceforth, we refer exclusively to stress fibers and cortical actin.

The sensor may exhibit varying degrees of localization to distinct F-actin architectures. To quantify this difference, we measure the density of the sensor in different F-actin structures, by the enrichment ratio (ρ). The enrichment ratio of the sensor is calculated by normalizing the fluorescence intensity of the RFP sensor to the fluorescence intensity of phalloidin actin. We then measured this value as a function of the radial distance from the center of a circular micropattern ($r$) to the outer radius of the micropattern ($R_0 = 12.5 \mu m$). Within this range, we segment the cell into the "central" area ($0 < r/R_0 < 0.6$) that contains cortical actin and SFs as well as the "peripheral" area ($1 > r/R_0 > 0.6$) that contains lamellipodial actin. We find that the sensor enrichment ratio (ρ) (Fig. 1g) was higher in the central regions than the peripheral regions, suggesting a low accumulation on branched F-actin and high accumulation in the lamella and inner region[59] (Fig. 1e, dashed line

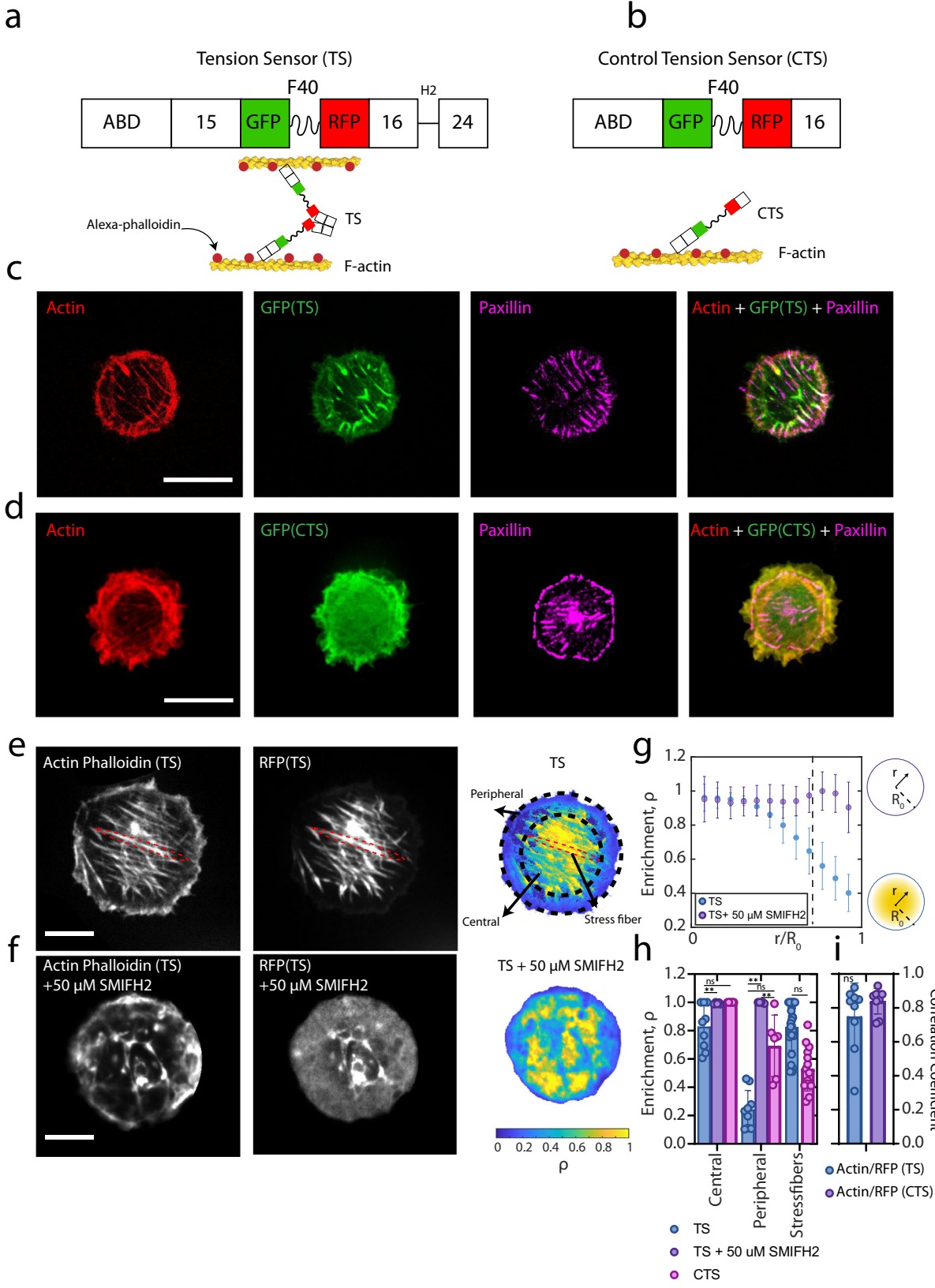

indicates $r = 0.6$). Further, within the central region, the enrichment ratio was higher within SFs than within cortical actin. Inhibition of SFs by 50 μM SMIFH2 (Methods) led to a more uniform distribution of the sensor throughout the cell (Fig. 1f), as the sensor became more localized to cortical and lamellipodial actin (Fig. 1f, h). In cellular regions

with low actin concentration, there could be artifacts of low expression affecting the calculation of the enrichment ratio. To test this possibility, we applied a fluorescence intensity threshold to the images to identify and exclude regions with less than 30% of the mean RFP intensity (Supplementary Fig. 4). Our measurements were robust to

**Fig. 1 | Tension sensor differentially localized to distinct F-actin structures.**
**a** Schematic of the tension sensor (TS), connected to actin filaments, and **b** control
tension sensor (CTS), connected to one actin filament. **c** TS expressing U2OS cell on
a circular micropattern of 0.2 mg/mL gelatin, showing phalloidin-stained actin, GFP
(TS), and paxillin. Scale bar is 25 μm. **d** CTS expressing U2OS cell on a circular
micropattern of 0.2 mg/mL gelatin, showing phalloidin-stained actin, GFP (CTS),
and paxillin. Scale bar is 25 μm. **e** Actin and TS GFP of U2OS cell on a circular
pattern. Red dash line indicates a stress fiber. Enrichment colormap showing the
normalized ratio of TS to actin. Scale bar is 10 μm. **f** Actin and TS GFP of U2OS cell
treated with 50 μM SMIFH2, on a circular pattern. Enrichment colormap showing
the normalized ratio of TS to actin. Scale bar is 10 μm. **g** Radially averaged sensor
enrichment (ρ) for TS and TS+SMIFH2 cells (*n* = 10 for TS cells, *n* = 13 for TS
+SMIFH2 treated cells). Dashed line indicates central-peripheral border. Data are

presentd as mean values ±SD. **h** Sensor enrichment (ρ) in the central, peripheral and
SF regions (For central and peripheral results, *n* = 10 for TS cells, *n* = 7 for CTS cells,
and *n* = 5 for TS+SMIFH2 cells. For stress fibers, *n* = 26 for TS, and *n* = 14 for CTS).
Two-way ANOVA was used for significance. (*p*(TS central and TS + SMIFH2 cen-
tral) = 0.006, *p*(TS central and CTS central) = 0.999, *p*(TS peripheral and TS +
SMIFH2 peripheral) = 0.006, *p*(TS peripheral and CTS peripheral) = 0.999, *p*(TS +
SMIFH2 peripheral and CTS peripheral) = 0.003, *p*(TS stress fibers and CTS stress
fibers) = 0.998. **i** iPearson colocalization coefficient for actin/RFP(TS) (*n* = 9) and
actin/RFP(CTS) (*n* = 9) cells. Two-sided t-test between the two samples *p*(TS and
CTS) = 0.2184. One sample two-sided *t*-test for each column *p* < 0.0001. Experi-
ments were repeated three times independently. Data are presented as mean values
±SD. Source data are provided as a Source Data file.

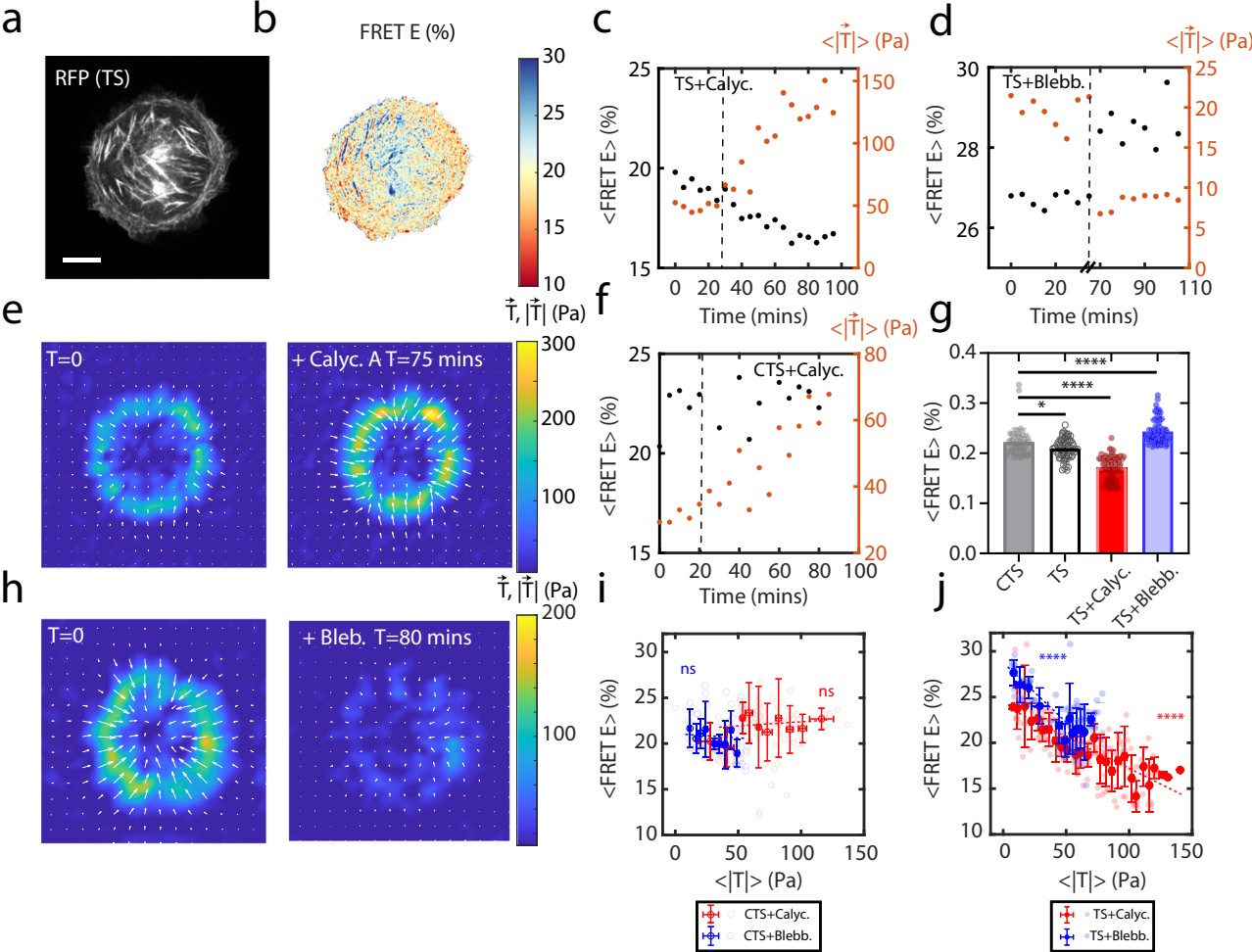

**Fig. 2 | Molecular tensions measured by the TS temporally correlate to trac-
tions from TFM. a** TS cell on a circular pattern, showing the RFP of the sensor
**b** and the associated FRET E map. Scale bar is 5 μm. Experiments were repeated
eight times independently. **c** mean FRET E (black dots) and traction (orange dots)
over time for a 10 nM Calycluin-A treated TS cell. **d** Traction magnitude and FRET E
over time for a 10 μM Blebbistatin-treated TS cell (**e**) corresponding traction vec-
tors (T̄) on the PAA gel before and after addition of Calyculin-A. Grid size is 6.4 μm.
**f** Traction and FRET E over time for a 10 nM Calyculin-A treated CTS cell. **g** FRET E
for TS, CTS and drug treated cells. CTS (*n* = 11 cells, *N* = 53 measurements at dif-
ferent times), for TS (*n* = 12 cells *N* = 77 measurements at different times) for TS
+Calyc. (*n* = 12 cells *N* = 87 measurements at different times), for TS+ Blebb. (*n* = 7
cells *N* = 87 measurements at different times). Tukey's test was used for

significance. Two-sided t-test with *p*(CTS and TS) = 0.0176, **** is *p* < 0.0001.
**h** corresponding traction vectors (T̄) on the PAA gel before and after addition of
Blebbistatin. Grid size is 6.4 μm. **i** <FRET E>-<|T̄|> relationship after 10 nM
Calyculin-A (*n* = 12, ns regression *p*-value = 0.1579) and 10 μM blebbistatin (*n* = 7, ns
regression *p*-value = 0.1192) treatment of CTS cell over time. Two-tailed t-test was
used for significance. ns is non-significant. Data are presented as mean values ±SD.
**j** <FRET E>-<|T̄|> relationship for Calyculin-A (*n* = 12 cells *N* = 77 measurements at
different times) and Blebb. (*n* = 7 cells *N* = 87 measurements at different times)
treated TS cells. Two-tailed t-test was used for statistical significance with
*p* < 0.0001 for each line. The slopes are significantly *p* < 0.0001 different
(*F* = 21.00). Data are presented as mean values ±SD. Source data are provided as a
Source Data file.

this threshold, and we therefore exclude artifacts as significantly affecting our calculations. Thus, we can conclude that the sensor localizes to all actin structures but is enriched within different structures to different extents.

As the sensor contains two ABDs, and functions as an F-actin crosslinker, we assume that the sensor reports tension between any two actin filaments to which it binds. To demonstrate that the sensor measures intracellular tension in-vivo, we next sought to explore potential correlations between molecular tensions measured by the TS and tractions exerted to the ECM. This approach tests the relationship between these variables but cannot be taken as a direct calibration between intracellular molecular tensions and external tractions.

### The tension sensor reports molecular tensions within F-actin

To assess the correlation of the molecular tensions within the F-actin cytoskeleton with traction forces, we compared the changes in mean FRET efficiency within cells (<FRET E>, Methods) to the magnitude of traction vectors <|$\vec{T}$|> transmitted to the ECM as measured by Traction Force Microscopy (TFM). In each case, cells adhere to circular micropatterns[57, 60] on 2.8 kPa gels ("Methods" section and Fig. 2a, b) such that all cells have the same spread area and same cell morphology. Henceforth, we use the term 'molecular tension' to describe the tensions across the TS measured by FRET, while the term 'traction' will be used to refer to the readouts obtained from TFM.

As described earlier, the FRET E is inversely related to the distance between RFP and GFP fluorophores. Thus, to establish the range of the sensor FRET Efficiency, we generate calibration controls, in which sensors have short or long domains between the fluorophores[61, 62]. To achieve high FRET E, the F40 linker in the TS construct was replaced with a short linker (GGSGGS)$_2$[30, 62] which showed FRET E≈33% ± 5%. Alternatively, a construct lacking the filamin sequences, composed solely of the fluorophores and the (GGSGGS)$_2$ linker was expressed as a soluble cytoplasmic protein. This construct did not localize with the F-actin and displayed a FRET efficiency of approximately ≈36% ± 4% (Supplementary Fig. 6). To achieve low FRET E, the F40 linker in the TS was replaced with a long linker (TRAF)[30, 62], which yielded FRET E≈12±4%. Also, a soluble cytoplasmic construct composed solely of the fluorophores and the TRAF linker was expressed with FRET E≈10% ±2 % ("Methods" section and Supplementary Fig. 6). It is noteworthy that the FRET E measurements were found to be statistically similar between the two methods of measuring high and low FRET E (Supplementary Fig. 6) and that the calibration linkers do not significantly extend in this context.

To determine if the measured FRET E pertains only to the stretching of the monomers (intramonomeric FRET), we assess the extent to which inter/intradimeric FRET occurs. Interdimeric FRET occurs when the GFP donor of one dimeric TS transfers energy to the RFP acceptor of another dimeric TS due to their proximity[63, 64]. Intradimeric FRET occurs within a single dimeric TS, when the GFP donor of one monomer transfers energy to the RFP acceptor of the other monomer. To account for these possibilities, we engineered mutants in which the GFP or the RFP were mutated to eliminate their fluorescence. These constructs were then expressed together at concentrations like standard FRET experiments. We verified that the inter/intradimeric FRET[63, 64] occurs at relatively low levels (6±2% at stress fibers, 11% ± 1% at the cortex and overall ≈13 ± 4%) falling within the lower range of FRET efficiency. We explain that the higher inter/intradimeric FRET at the cortex compared to the SFs (Supplementary Fig. 7), and other sensors[62, 64] may be due to the sensor geometry, and how TS dimers bind and accumulate on different F-actin architectures. For example, it may be possible that the monomers within a single TS construct bind in closer proximity at the cortex compared to stress fibers, potentially resulting in increased intradimeric FRET (Supplementary Fig. 16)[63]. Nevertheless, as interdimeric FRET is a function of sensor expression levels[30, 65], showing that baseline FRET efficiencies

are not significantly regulated by the TS expression levels (Supplementary Fig. 9), suggests interdimeric FRET is not obscuring the FRET signal. Therefore, we suggest that the elevated FRET efficiency observed at the cortical actin (11%±1%) is likely attributable to the intradimeric FRET. In any event, we have considered potential effects that the inter/intradimeric FRET may have on our FRET E measurements (Supplementary Fig. 7 and "Methods" section).

Over time, cells spread on adhesive micropatterns showed a constant average of traction magnitude <|$\vec{T}$|> and FRET E (Supplementary Fig. 8 and Supplementary Video 1). To probe the relationship between FRET E and tractions, we use activators and inhibitors of myosin activity. Upon treatment with 10 nM Calyculin-A, a myosin activator[66] (added at ≈ 00:30 (dashed line), Fig. 2c, e and Supplementary Video 2), the FRET E decreased and <|$\vec{T}$|> increased[67] (Fig. 2f and Supplementary Fig. 8). By contrast, upon treating the TS cells with 10 μM Blebbistatin, a myosin inhibitor, FRET E increased, and <|$\vec{T}$|> decreased (Fig. 2d, h and Supplementary Video 3). Thus, we recovered the inverse relationship between tractions and FRET E. However, it should be noted that imaging was performed with 488 nm light, which inhibits Blebbistatin[68]. Thus, to ensure that myosin remained inhibited, we incubated the cells with Blebbistatin for 30 min prior to imaging and limited our exposures to 300 ms every 5 min. In both cases, there is an inverse correlation ($p$< 0.001) between the FRET E and <|$\vec{T}$|> (Supplementary Fig. 8). By contrast, CTS cells did not show a significant change in the FRET E after 10 nM Calyculin-A or 10 μM Blebbistatin treatment (Fig. 1i, Supplementary Fig. 8). We also note that Ig16 has potential binding interactions with certain intracellular components[69], which may invalidate the assumption that the CTS functions as a load-free control. To address this, we conducted FRET E measurement on lysed cells, in which the molecular tension sensing module is diffuse, and thus experiences no molecular tension (Methods). We found that FRET E values for TS and CTS in cell lysates were similar to values in intact CTS cells (i.e., no significant differences, Supplementary Fig. 8). These results confirmed that the CTS construct is not affected by myosin-dependent contractility and the potential interaction of Ig16 with intracellular structures[69] is negligible in our analysis. Thus, we observe inverse correlations between the spatially averaged FRET E and <|$\vec{T}$|> (Fig. 2g, j and Supplementary Fig. 8). An inverse correlation between FRET E and molecular tension was also shown in single-molecule experiments for other molecular tension sensors[22, 34]. It is important to highlight that the observed correlation between molecular tensions and tractions does not represent a quantitative calibration between internal and external forces as the probed molecular tension between actin filaments may not be directly translated into tractions reflecting the entire cytoskeletal structure. This may be due to the tension sensor's dimerized geometry, mechanical coupling to the ECM, or viscous dissipations. Nevertheless, the correlation serves as an indication of the tension sensor's functionality in measuring inter-filament tensions in vivo and its sensitivity to forces generated by myosin.

The FRET efficiency of the sensor spans a range from molecular tension (FRET E ≈15%) to molecular compression (FRET E ≈33%). The transition from tension to compression occurs at the rest length of the F40 linker with approximately ≈22±2% corresponding to the FRET E of the load-free CTS, which appears to be comparable to the FRET E of the TS (≈24%±2%) at <|$\vec{T}$|>≈0. Thus, depending on the value of FRET E, an increase in FRET E can result from two distinct factors: the fluorophores moving closer together than the rest length of the F40 linker, as well as an elevated level of inter/intradimeric FRET. In this case, the actin filaments may be undergoing compression as they move closer together. Conversely, a decrease in FRET E indicates an increase in distance between filaments, implying tension. Although the FRET E of the sensing module (F40) has been previously calibrated for tension by optical tweezers[22], we use FRET E as the reporter of both molecular tension and compression. It should be noted that the presence of

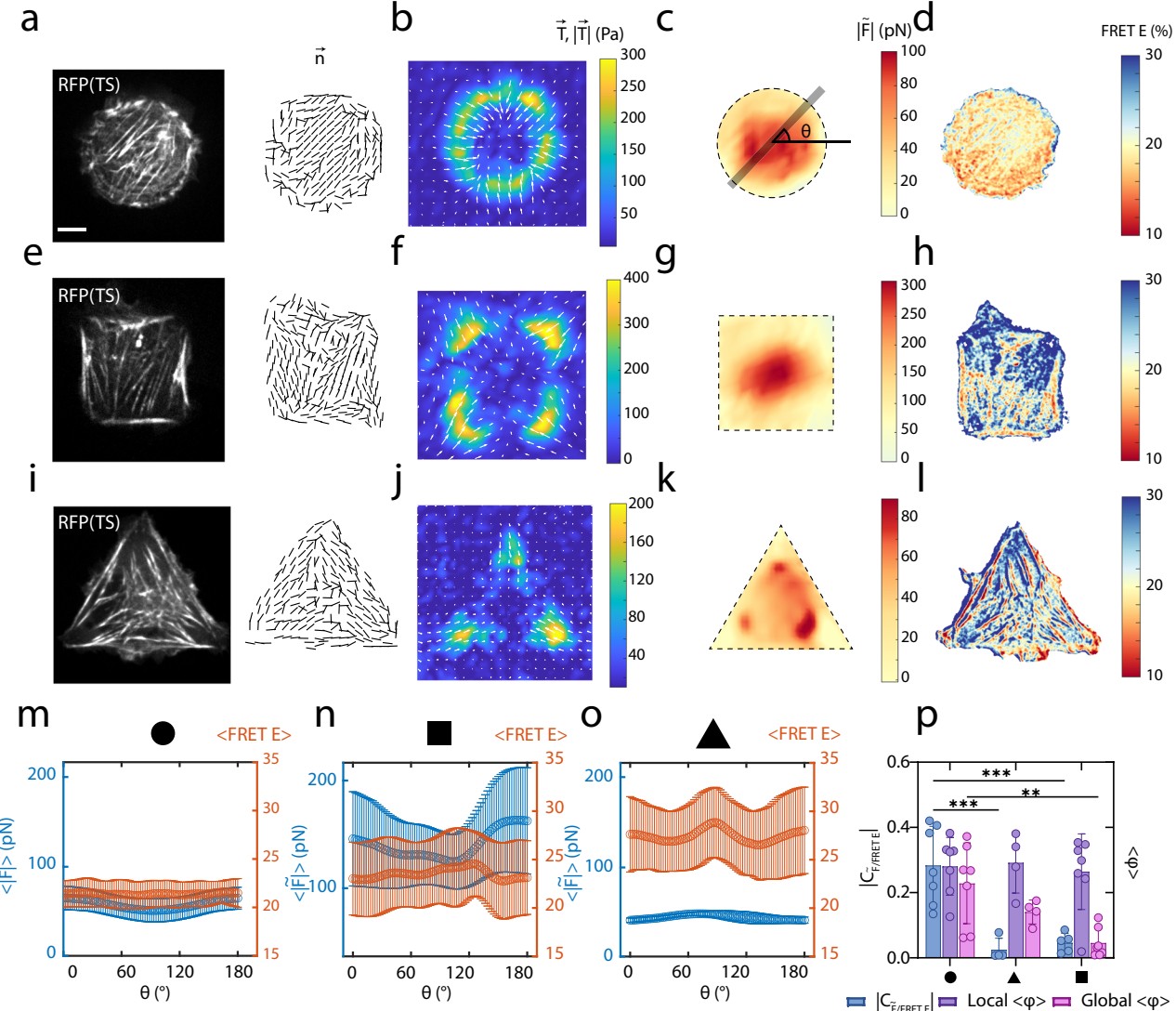

**Fig. 3 | Elastic assumptions fail to predict intracellular forces. a** RFP of a circular U2OS TS cell and the corresponding alignment field $\vec{n}$. **b** The magnitude and vectors of tractions. Grid size is 6 μm. **c** The average internal force $|\tilde{F}|$ calculated from the TFM tractions based on linear elasticity and force balance. **d** The FRET E of circular cell. **e** RFP of a square U2OS TS cell and the corresponding alignment field $\vec{n}$. **f** The magnitude and vectors of tractions. Grid size is 6.4 μm. **g** The average internal force $|\tilde{F}|$. **h** The FRET E of circular cell. **i** RFP of a triangle U2OS cell and the corresponding alignment field $\vec{n}$. **j** The magnitude and vectors of tractions. Grid size is 6.4 μm. **k** The average internal force $|\tilde{F}|$. **l** The FRET E of circular cell. **m** Radially averaged and interpolated $|\tilde{F}|$ (blue dots) and <FRET> (orange dots) as a

function of $\theta$, the angle from the center of circle, (**n**) square and (**o**) triangle shapes. **p** $C_{|\tilde{F}|/\text{FRET E}}$, local <$\varphi$> and global <$\varphi$> for circle, square, and triangle shapes. ($n = 7$ for circles, $n = 6$ for square and $n = 5$ for triangles). Two-way ANOVA test was used for significance for $C_{|\tilde{F}|/\text{FRET}}$ $p$(circle and triangle) = 0.0001, $p$(circle and square) = 0.0002, $p$(triangle and square) = 0.9345. For local <$\varphi$> $p$(circle and triangle) = 0.9795, $p$(circle and square) = 0.9380, $p$(triangle and square) = 0.8790. For global <$\varphi$>, $p$(circle and triangle) = 0.2807, $p$(circle and square) = 0.0014, $p$(triangle and square) = 0.2293. In all panels, data are presented as mean values ±SD. Scale bar is 5 μm. Source data are provided as a Source Data file.

intradimeric or interdimeric could potentially render the previous calibrations invalid for this investigation. We alternatively quantify the FRET signal to get the FRET index, which gives comparable results to FRET E (Supplementary Fig. 9), however, due to higher sensitivity of the FRET index to the expression level of the sensor (Supplementary Fig. 9), we report our results in terms of FRET E.

In comparing tractions to FRET E, we note that at low tractions, Blebbistatin-treated cells exhibit significantly higher FRET E compared to the FRET E of CTS, lysed TS, and lysed CTS cells (Supplementary Fig. 8). Based on the calibration controls, a higher FRET E is suggestive of an average compression response at the molecular level (Fig. 2g). However, Blebbistatin and Calyculin-A treatments exhibited different <$\vec{T}$>-FRET E slopes, indicating that the tractions and molecular tensions do not have a unique relationship across separate experiments.

This could be due to differences in inter/intradimeric FRET within distinct actin architectures induced by various drugs, F-actin content, cell-ECM coupling[70, 71], or cell stiffness[72]. Furthermore, it should be noted that alternative myosin inhibitory drugs such as Y27632 or ML7 were not used in this assay, as they were not as compatible with our micropatterning protocol. With these controls, cells were poorly confined to patterned areas[73] (Supplementary Fig. 2).

## Molecular tensions highlight the limitation of elastic assumptions in measuring intracellular forces

Assuming homogenous-isotropic elasticity (HIE), internal forces ($|\tilde{F}|$) (Fig. 3c, g, k) were calculated by a force balance analysis on ($\vec{T}$)[74] (Supplementary Note 1). For cells with a circular shape, $|\tilde{F}|$

exhibited similar distributions to the FRET E (Fig. 3a–d). In this case, traction is peripheral, and force balance yields an approximately uniform distribution of force with higher forces at the center than at the periphery. By contrast, for cells with a square or triangular shape, $|\widetilde{F}|$ showed a dissimilar localization to the FRET E. In these cases, the traction was highest at the edges (Fig. 3e–l), where the SFs were present, which differs from the distribution of $|\widetilde{F}|$ (Fig. 3g, k). Radial averages of $|\widetilde{F}|$ and FRET E for circles showed a homogenous distribution of the FRET E (Fig. 3m), whereas in squares and triangles, the FRET distribution was heterogenous (Figs. 3n, 3o). The correlation between radial averages of $|\widetilde{F}|$ and FRET E was quantified by a spatial Pearson analysis ($|C_{|\widetilde{F}|/FRET}|$) (blue bars, Fig. 3p and "Methods" section), showing significantly higher correlation coefficients in circles than other shapes. We further verified that the FRET index, like FRET E consistently exhibits spatially uniform distributions in circular shapes, and spatially non-uniform distributions in squares and triangles (Supplementary Fig. 10).

The different spatial distributions of FRET E and $|\widetilde{F}|$ are explainable by the complex transmission of the molecular tensions to tractions, possibly due to the differences in cytoskeletal organization and how these organizations bear mechanical loads. Therefore, we quantify the cytoskeletal structure within the cells on different patterned shapes (e.g. net alignment of SFs). In circular cells, SFs are often distributed across the central area nearly uniformly (in which $|\widetilde{F}|$ and FRET E correlate well), while they are predominantly peripheral in squares (in which $|\widetilde{F}|$ and FRET E poorly correlate). We measure the mean nematic alignment order parameter, $\langle \varphi \rangle$ (Methods) of F-actin images for circular and non-circular shapes. The order parameter quantifies the alignment of SFs, in which $\langle \varphi \rangle$ is calculated across a radius $r$ (Methods). The value of $\langle \varphi \rangle$ varies between 0 (unaligned) and 1 (aligned). We find that the 'local' order parameter with $r = 5 \, \mu m$ is similar across different shapes (Fig. 3p). Upon measuring the 'global' nematic order with $r = 10 \, \mu m$, cells on circular patterns exhibited significantly higher order compared to cells on triangle and square patterns (Fig. 3p). Furthermore, the spatial correlation analysis (Methods) between the FRET E and the global order $\langle \varphi \rangle$ ($|C_{\varphi/FRET}|$) showed an inverse correlation in circular cells, as opposed to no correlation or a positive correlation in square or triangle shapes (Supplementary Fig. 11). Moreover, the presence of perpendicular or angled arrangement of SFs in square or triangle patterns may underlie the differences between $|\widetilde{F}|$ and FRET E. Therefore, to explore the effect of SFs alignment on the intracellular molecular tension, we controlled the SFs alignment and performed mechanical stretching tests on the cells.

### Mechanical anisotropy of the cell is strain-dependent

Next, we examined the role of SFs in establishing mechanical anisotropy during uniaxial stretching of the cells[29,39,75]. To do so, we seeded cells onto thin PDMS films, that contain microgrooves of 500 nm height and width (Fig. 4a and "Methods" section). The cells that adhere to the microgrooves will elongate and align their shape along the directions of the grooves[76]. This enables the strain ($\varepsilon_{applied}$) to be applied parallel (∥) to the direction of cell alignment ('parallel' cells) or orthogonal (⊥) to the direction of cell alignment ('orthogonal' cells). Further, the PDMS film has a Poisson ratio of ≈ 0.5. Thus, stretching the film stretches the cells in one direction but compresses them in the orthogonal direction.

A square wave strain with a ramp time of 5 seconds was applied to the substrate (Supplementary Fig. 12) in two consecutive cycles. The strain ramp time was set to be shorter than the tension sensor turnover time of ≈20 s, i.e. a measure of the time it takes for bound sensors to unbind and diffuse sensors to bind. The turnover time was calculated using photobleaching through FRAPPA and recovery of the RFP and GFP ("Methods" section and Supplementary Fig. 15). To calculate the

net change in the molecular tension due to stretching, we calculated ΔFRET E, using the following equation:

$$\Delta \text{FRETE} = \frac{<FRETE_s> - <FRETE_r>}{<FRETE_r>} \quad (1)$$

with $<FRET \, E_s>$ and $<FRET \, E_r>$ indicating the FRET efficiency of stretched and relaxed states, respectively. Therefore, ΔFRET E is negative for net tension and positive for net relaxation/compression. Also, a ΔFRET E ≈0 is indicative of no resistance to stretch.

Stretching parallel TS cells (Fig. 4b) by $\varepsilon_{applied} = 5\%$ or 10% did not significantly change the FRET E. However, at 15% and 20% strains the ΔFRET E decreased by ≈0.05±0.03 and ≈0.08±0.04, respectively. (Fig. 4c). By contrast, orthogonal cells showed a positive ΔFRET E in the cytoskeleton at 5% strain and insignificant ΔFRET E at the higher strains (Fig. 4g). The CTS cells, however, did not show a significant change in ΔFRET E (Figs. 4d and 4h). The positive ΔFRET E indicates relaxation (decrease in tension) rather than compression, as we see the FRET E in TS cells (≈14%±1.8%) stays below the FRET E in CTS cells (E≈22%±1.7%, Supplementary Fig. 12). Moreover, compared to TS cells on circular 2.8 kPa gels, the FRET E of TS cells on PDMS substrates tends to be lower. This may be due to a higher stiffness of PDMS[77], or polarization of the cytoskeleton[4] induced by the nano-grooved substrate. As the sign of the ΔFRET E changes depending on the direction of applied strain, we term the tension-strain response as 'anisotropic'.

To explore the role of myosin II generated molecular tension[23,27] in the observed anisotropy, cells were treated with 10 μM Y27632– a ROCK inhibitor[78] (Fig. 4b, e, i) or 67 μM ML-7[7] – a myosin light chain kinase (MLCK) inhibitor (Fig. 4b, f, j). No resistance to stretch was observed in parallel cells under Y27632 treatment cells (ΔFRET E≈0.009 ± 0.06 at 15% strain, and 0.009 ± 0.04 at 20% strain), as compared to non-treated cells. Similarly, under ML-7 treatment all ΔFRET E are attenuated (≈0.002±0.03, ≈0.006±0.02, ≈−0.02±0.03, ≈−0.008±0.03, at 5%, 10%, 15%, and 20% strains), as the tension sensor do not exhibit changes in net tension (Fig. 4f). The absence of measurable FRET changes at high strains compared to no strain in the presence of myosin inhibitors suggest that the TS does not report all tensions. Inhibited myosin activity may impact SF elasticity[6,79], disassembly, and integration[80,81] with other F-actin structures, such that molecular tensions fall below a measurable threshold value.

Under Y27632 treatment for orthogonal cells, however, a tensile response was observed at 5% strain which attenuates at larger strains (Fig. 4i), consistent with ML-7 treatment (Fig. 4j). The shift from relaxation/compression to tension upon inhibition of myosin may be attributed to the inactivation of stretch-induced myosin contraction[82,83] or the binding of myosin to actin in response to stretch[84], which counteracts the increase in molecular tension. We alternatively inhibited myosin activity with 10 and 20 μM Blebbistatin, which showed a dose-dependent and relaxation response at high strains (Supplementary Fig. 13). However, cells detached from the substrate at a high 20% strain, when treated with 20 μM Blebbistatin, suggesting disruptions to the cytoskeleton or cell-ECM couplings. As Blebbistatin is known to directly perturb the F-actin cytoskeleton[78] and myosin's actin-binding affinity[85], we expected disruptions to the cytoskeleton at high strains. However, in the absence of external strains, we measured the Blebbistatin-induced changes in molecular tensions and tractions as shown in the previous section, despite possible cytoskeletal perturbations.

Upon inhibition of the Arp2/3 complex via 200 μM CK666 which disrupts the branched actin composition in the cortex[86], we did not observe a significant change in ΔFRET E for orthogonal cells (ΔFRET E ≈−0.015±0.04, ≈0.02±0.06, ≈−0.02±0.05, ≈0.03±0.05, at 5%, 10%, 15%, and 20% strains) as compared to non-treated cells which exhibited significant changes. However, in the parallel direction, cells showed a consistent FRET response compared to non-treated cells

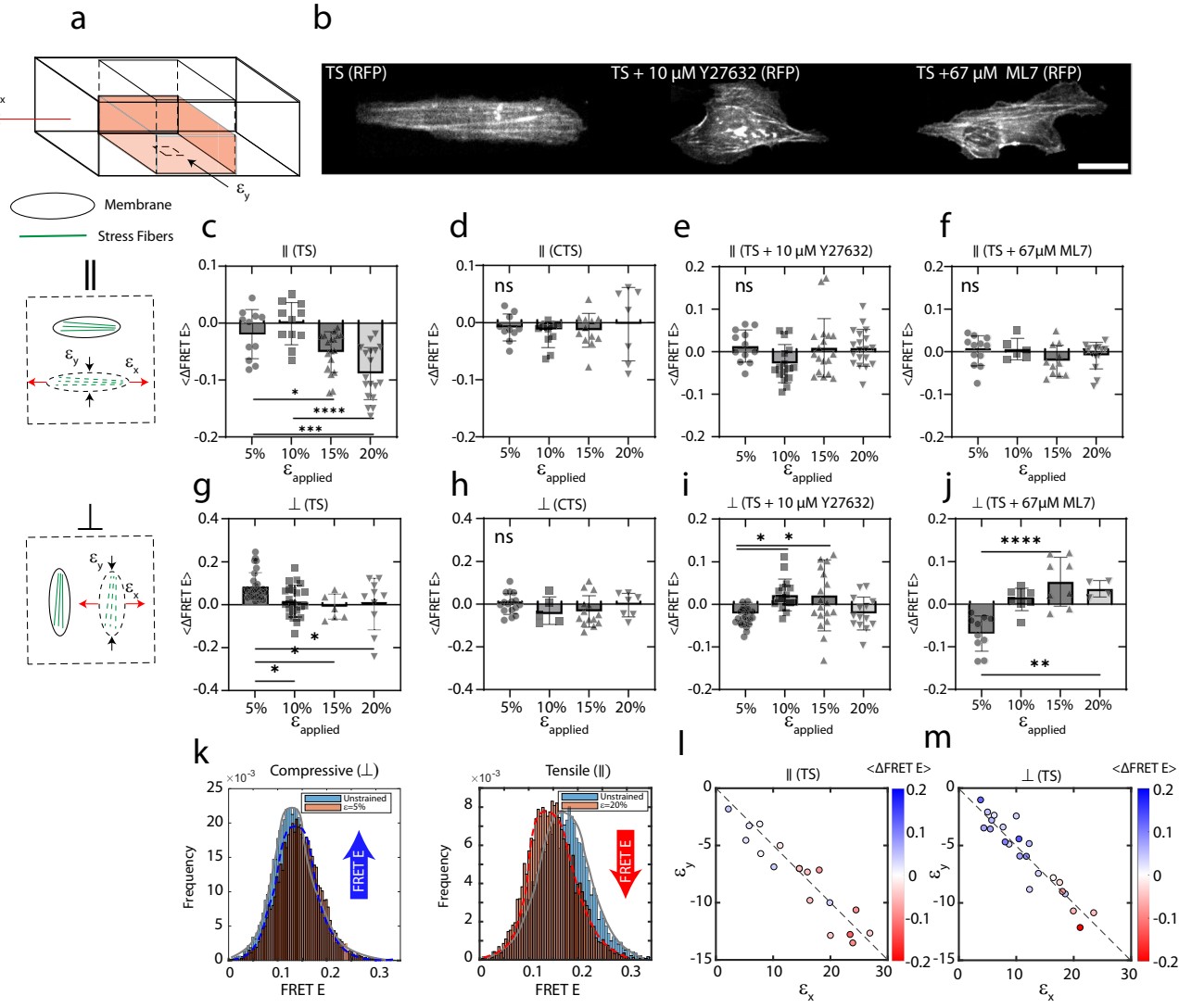

**Fig. 4 | Cell mechanical anisotropy is strain dependent. a** schematic of stretching chamber, cells and their SFs aligned parallel (∥) and orthogonal (⊥) to the direction of stretch. $\varepsilon_x$ indicated the applied strain, and $\varepsilon_y$ the transverse compression from Poisson effect. **b** Images in RFP channel of TS, Y27632 treated and ML7 treated cells. Scale bar is 20 μm. **c** ΔFRET E for TS parallel cells (∥), with 5%-20% applied strains. ($n = 11$ for 5%, $n = 11$, for 10%, $n = 18$ for 15% and $n = 18$ for 20% strains). One-way ANOVA was used for significance test. $p$(5% and 10%) = 0.7156, $p$(5% and 15%) = 0.1266, $p$(5% and 20%) = 0.0001, $p$(10% and 15%) = 0.0101, $p$(15% and 20%) = 0.0318. **d** ΔFRET E for parallel CTS cells (∥). $n = 10$ for 5%, $n = 10$, for 10%, $n = 12$ for 15% and $n = 7$ for 20% strains. One-way ANOVA statistical significance with $p$-value = 0.7228 **e** ΔFRET E for Y27632 treated parallel TS cells (∥). One-way ANOVA statistical significance with $p$-value = 0.0877 ($n = 12$ for 5%, $n = 16$, for 10%, $n = 18$ for 15% and $n = 19$ for 20% strains). **f** ΔFRET E for ML7 treated parallel TS cells (∥)($n = 10$ for 5%, $n = 8$ for 10%, $n = 8$ for 15% and $n = 4$ for 20% strains). One-way ANOVA statistical significance with $p$-value = 0.2683. **g** ΔFRET E for orthogonal TS cells (⊥). ($n = 29$ for 5%, $n = 22$, for 10%, $n = 7$ for 15% and $n = 20$ for 20% strains). One-way ANOVA was used for significance test. $p$(5% and 10%) = 0.0142, $p$(5% and

15%) = 0.0278, $p$(5% and 20%) = 0.0268, $p$(10% and 15%) = 0.8749, $p$(10% and 20%) = 0.9671, $p$(15% and 20%) = 0.9899. **h** ΔFRET E for orthogonal CTS cells (⊥) ($n = 16$ for 5%, $n = 5$, for 10%, $n = 14$ for 15% and $n = 7$ for 20% strains). One-way ANOVA statistical significance with $p$-value = 0.4127. **i** ΔFRET E for Y27632 treated orthogonal TS (⊥) ($n = 34$ for 5%, $n = 16$, for 10%, $n = 19$ for 15% and $n = 14$ for 20% strains). One-way ANOVA test was used for significance test. $p$(5% and 10%) = 0.0219, $p$(5% and 15%) = 0.0160, $p$(5% and 20%)>0.99, $p$(10% and 15%)>0.99, $p$(10% and 20%) = 0.0772, $p$(15% and 20%) = 0.0690. **j** ΔFRET for ML7 treated orthogonal TS cells. $n = 10$ for 5%, $n = 8$, for 10%, $n = 7$ for 15% and $n = 4$ for 20% strains. One-way ANOVA was used for significance test. $p$(5% and 10%) = 0.0019, $p$(5% and 15%) < 0.0001, $p$(5% and 20%) = 0.0012, $p$(10% and 15%) = 0.2132, $p$(10% and 20%) = 0.7651, $p$(15% and 20%) = 0.9110. **k** FRET E histogram for cells undergoing relaxation and tensile response at 5% and 20% strains. **l** ΔFRET E as a function of $\varepsilon_x$ and $\varepsilon_y$ for parallel TS cells (∥). **m** ΔFRET E as a function of $\varepsilon_x$ and $\varepsilon_y$ for orthogonal TS cells (⊥). In all panels, data are presented as mean values ±SD. Source data are provided as a Source Data file.

(Supplementary Fig. 13). Comparing the distribution of the FRET E values in unstrained vs. strained cells (<20% strain) showed a uniform increase or decrease (Fig. 4k) in relaxation or tensile response, respectively. Therefore, we established that at 5-20% applied strains, the tension sensor predominantly captures molecular relaxation or tension (Fig. 3l, m). Baseline FRET E values indicated that the TS is not completely unloaded under different inhibitory conditions (Supplementary Fig. 12). The TS in cells under Inhibitory drugs exhibits a

baseline tension, which undergoes relaxation or increased tension depending on the direction or magnitude of applied strain. We also notice the differences in FRET are more significant by calculation of relative FRET changes. We explain the reason for the difference in significance due to the intercellular variability in the FRET efficiencies as they are not adjusted for the relaxed FRET values.

Strain-induced changes in the FRET E may also be due to the unbinding/binding of the sensor or the appearance of cytosolic

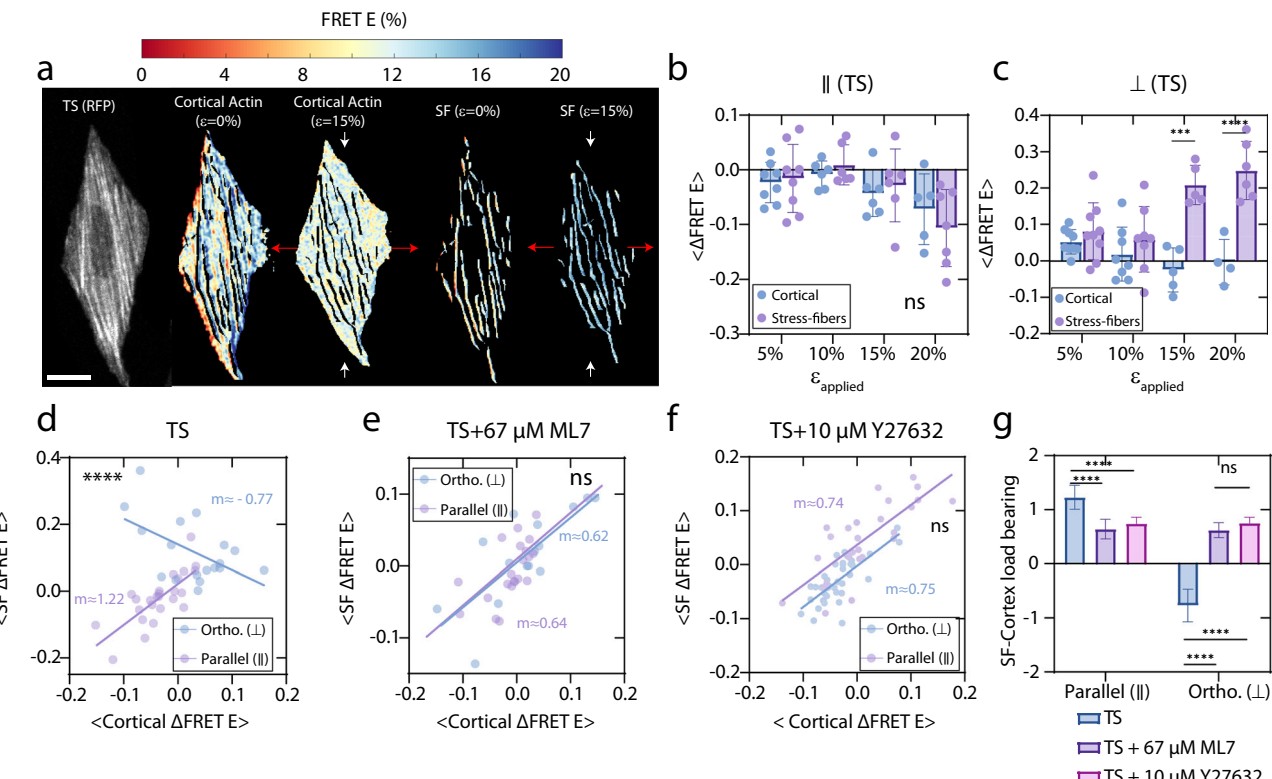

**Fig. 5 | Stress fibers and cortex bear distinct mechanical loads. a** from left to right: orthogonal U2OS cell aligned (⊥) (left), cortical actin at 0% strain, cortical actin at 15% strain, SFs at 0% strain, SFs at 15% strain. **b** ΔFRET E of SFs and cortical actin in parallel TS cells (||) under strain. Data are presented as mean values ±SD. $n = 8$ for 5%, $n = 8$ for 10%, $n = 5$ for 15%, and $n = 6$ for 20% strain. Šidák's significance test for 5% $p$(SFs and cortical actin) = 0.9983, for 10% $p$(SFs and cortical actin) = 0.6172, for 15% $p$(SFs and cortical actin) = 0.9869, for 20% $p$(SFs and cortical actin) = 0.9937. **c** ΔFRET E of SFs and cortical actin in orthogonal TS cells (⊥) under strain. $n = 9$ for 5%, $n = 8$ for 10%, $n = 5$ for 15%, and $n = 6$ for 20% strain. Data are presented as mean values ±SD. Šidák's significance test for 5% $p$(SFs and cortical actin) = 0.9943, for 10% $p$(SFs and cortical actin) = 0.7562, for 15% $p$(SFs and cortical actin) = 0.0013, for 20% $p$(SFs and cortical actin)<0.0001. **d** SF ΔFRET E against cortical ΔFRET E for TS cells. Two-sided $t$-test significance for the slope of the lines, $p$(ortho) = 0.0213, $p$(parallel)<0.0001. The slopes difference $p$-value < 0.0001, $F = 28.37$. **e** SF ΔFRET E as a function of cortical ΔFRET E for 67 µM ML7

treated cells. Two-tailed $t$-test significance for the slopes, $p$(ortho) = 0.0024, $p$(parallel) = 0.0009. The slopes difference $p$-value = 0.9355, $F = 0.0066$. **f** SF ΔFRET E against ΔFRET E for TS+10 µM Y27632 cells. Two-sided $t$-test significance for the slopes, $p$(ortho) < 0.0001, $p$(parallel) < 0.0001. The slopes difference $p$-value = 0.9266, $F = 0.008561$. **g** SF-cortex load bearing in parallel and orthogonal directions, compared to ML7 and Blebbistatin-treated cells. Data are presented as the slope of the linear regression, ± regression SD. Šidák's significance test for parallel cells, $p$(TS and TS + 67 µM ML7) < 0.0001, (TS and TS + 10 µM Y27632) < 0.0001, (10 µM Y27632 and TS + 67 µM ML7) = 0.9852. Šidák's significance test for orthogonal cells, $p$(TS and TS + 67 µM ML7) < 0.0001, (TS and TS + 10 µM Y27632) < 0.0001, (10 µM Y27632 and TS + 67 µM ML7) = 0.8807. Error bars are SD of regression lines. $n = 15$ for ⊥(TS), $n = 13$ for || (TS), $n = 14$ for ⊥ (TS +ML7), $n = 19$ for || (TS+ML7), $n = 11$ for ⊥(TS+Blebb.), $n = 9$ for ||(TS+Blebb.). Scale bar is 5 µm. Source data are provided as a Source Data file.

sensors in the focal plane. The quantification of the binding/unbinding of the sensor due to applied strain was performed by measuring the total amount of the sensor ΔRFP and the density of the sensor ΔRFP/A, where A is the cell area. With ΔRFP≈−0.05, we verified that in orthogonal cells, despite a basal level of sensor unbinding, the sensor remains bound during stretching (Supplementary Fig. 12). Decreasing ΔRFP/A (≈−0.10, ≈−0.16, ≈−0.21, ≈−0.31 for at 5%, 10%, 15%, and 20% strain), further verified that the density of the sensor decreases with strain, as the area increases with strain. However, in parallel cells there is an increase in the density (ΔRFP/A≈0.2) and the total amount of the sensor (ΔRFP ≈0.2), suggesting a deformation-induced revealing of binding sites on actin or integrins[54,87] (Supplementary Fig. 12). As this effect is not observed in the orthogonal direction, we suggest the elongation of stress fibers results in more recruitment of the sensor.

## Applied strain alters how mechanical tension is shared between cortical actin and stress fibers

Next, we sought to understand how molecular tension changes in SFs versus cortical actin during cell stretch. Thus, we segmented the fluorescence images into distinct regions, which correspond to SFs

and cortical actin separately (Fig. 5a, Supplementary Fig. 14 and Methods). To do so, we generated a binary F-actin mask[88] for the FRET E for the SFs and an inverted mask for the cortical region. In parallel cells, the ΔFRET E of cortical actin and SFs decrease, undergoing net molecular tensions (Fig. 5b). The decrease in FRET E can also occur due to a decrease in intradimeric FRET between dimers of TS. In stress fiber sliding under shear forces TS molecules move away from each and likely change their configuration (Supplementary Fig. 16C). On the cortical actin, normal and shear forces likely increase the distance between dimeric TS sensors (interdimeric FRET), as well as increasing tension through intradimeric and intramonomeric FRET (Supplementary Fig. 16A). Conversely, in orthogonal cells, ΔFRET E of the SFs increase, while decreasing for the cortical actin (Fig. 5c). The increase in FRET E of the SFs is a result of their relaxation, as well as a potential increase in the interdimeric FRET by relaxation of the dimers getting closer together (Supplementary Fig. 16 B). Although there is a larger level of inter/intradimeric FRET in cortical actin, we observed a reduction in FRET efficiency. This is potentially a consequence of increased tension and a decrease in inter/intradimeric FRET in the stretched cortex as the dimers of TS move away under stretch (Supplementary Fig. 16A). Again, the different change in ΔFRET E is

indicative of mechanical anisotropy of the cell and differential load bearing in SFs and the cortical actin.

We then plot the ΔFRET E for SFs versus the ΔFRET E for cortical actin. The slope of the plot (*m*) measures the extent of accumulation (m>0) or relaxation (m<0) of molecular tensions in SFs compared to the cortical actin. In the orthogonal direction, ΔFRET E of SFs showed a significant correlation to the ΔFRET E of cortical actin, for both compressive and tensile responses. In this case, $m \approx -0.77$ (m<1, *p*-value < 0.001) (Fig. 5d). In the parallel direction, cells similarly showed a strong correlation between the ΔFRET of SFs and cortical actin, with $m \approx 1.22$ (m>1, *t*-test *p*-value < 0.001) (Fig. 5d). Higher ΔFRET E of SFs is suggestive of higher accumulation of molecular tensions in SFs compared to the cortical actin. This can also be reflective of how inter/intradimeric FRET changes in these compartments (Supplementary Fig. 16). The slopes were significantly different ($P < 0.0001$), suggesting an anisotropic mechanical response, and differential load bearing between SFs and the cortical actin.

To evaluate the influence of myosin-generated forces and the inter/intradimeric FRET in our measurements, we conducted additional analyses using ML7 and Y27632 drugs while retaining the presence of the cortex and SFs. Upon inhibition of myosin II activity using 67 μM ML7 (Fig. 5e) or 10 μM Y27632 (Fig. 5f), SFs were still present. Again, a strong correlation was observed for ML7 (m≈0.64 and m≈0.62 in parallel or orthogonal directions) or Y27632 (m≈0.74 and m≈0.75 in parallel or orthogonal directions) -treated cells. However, the difference in the slopes was nonsignificant in two directions. Assuming inter/intradimeric FRET is consistent between treated and non-treated SFs and cortical networks, our results indicate that in the absence of myosin activity, SFs accumulate less molecular tension compared to cortical actin. These results indicated that SFs and cortical regions exhibit differential load bearing depending on the direction of strain (Fig. 5g).

## Discussion

Here, we present the development of a FRET sensor to report on molecular tension within the F-actin cytoskeleton in space and time. With spatial localization, we attribute local changes in the FRET E to local changes in molecular tension within SFs and the F-actin cortex. Using this capability, we show how different F-actin structures bear different mechanical loads, and that the balance depends upon cell morphology, the magnitude of the applied strain, and the direction of the applied mechanical load.

First, we note that the tension sensor is not functionally equivalent to a FLMN-A crosslinker. Rather, we used the actin-binding domain (ABD) of FLMN-A to develop a sensor that binds to both cortical actin and stress fibers and reports molecular tensions between actin filaments within these structures. As the TS binds to two actin filaments within bundled and branched structures, the measured FRET E may indicate the magnitude of molecular tensions. However, the TS may not capture all internal molecular tension in the cytoskeleton. These molecular tensions arise from forces acting either perpendicular or parallel to the direction of the actin filaments, representing the magnitude of effective forces. The parallel forces include resistance force to filament sliding within stress fibers, or axial forces along a single actin filament. Nevertheless, due to the dimerized geometry of the sensor, distinguishing the directions of these forces within distinct cytoskeletal architectures is challenging. The geometry of the TS may also impact the range of FRET efficiencies and inter/intradimeric FRET on distinct actin architectures, which may be resulting in relatively higher inter/intradimeric FRET in the cortex compared to stress fibers. Consequently, this explains why inter/intradimeric FRET appears higher when compared to previously developed tension sensors featuring simpler geometries[34, 62, 64].

The existence of different slopes in the correlation between molecular tensions (FRET E) and tractions suggests that there is no unique molecular tension-traction relationship across different scales. With Blebbistatin, we suggest that its ability to hinder force generation and crosslinking can alter the cytoskeletal stiffness[78], leading to weaker molecular tensions. Also, the cell-ECM mechanical coupling[71] can influence the transmission of molecular tensions to the ECM tractions and thus a mismatch between molecular tensions and tractions.

Previously it has been demonstrated that vinculin undergoes tension at the edges of isotropic (e.g., circular) and anisotropic (e.g. square) shapes[60], similar to the F-actin tension reported by the TS. However, the TS indicates compression solely in anisotropic shapes, whereas the vinculin tension sensor reports compression regardless of the cell shape[60]. We then demonstrate that force balance on TFM with elastic assumptions, and molecular tensions do not always result in similar distributions. While isotropic shapes showed similarity between the FRET E distribution and the inferred internal forces $|\widetilde{F}|$, this correlation is diminished for anisotropic shapes. Thus, lack of consideration for the internal configuration of F-actin by the force balance, along with the

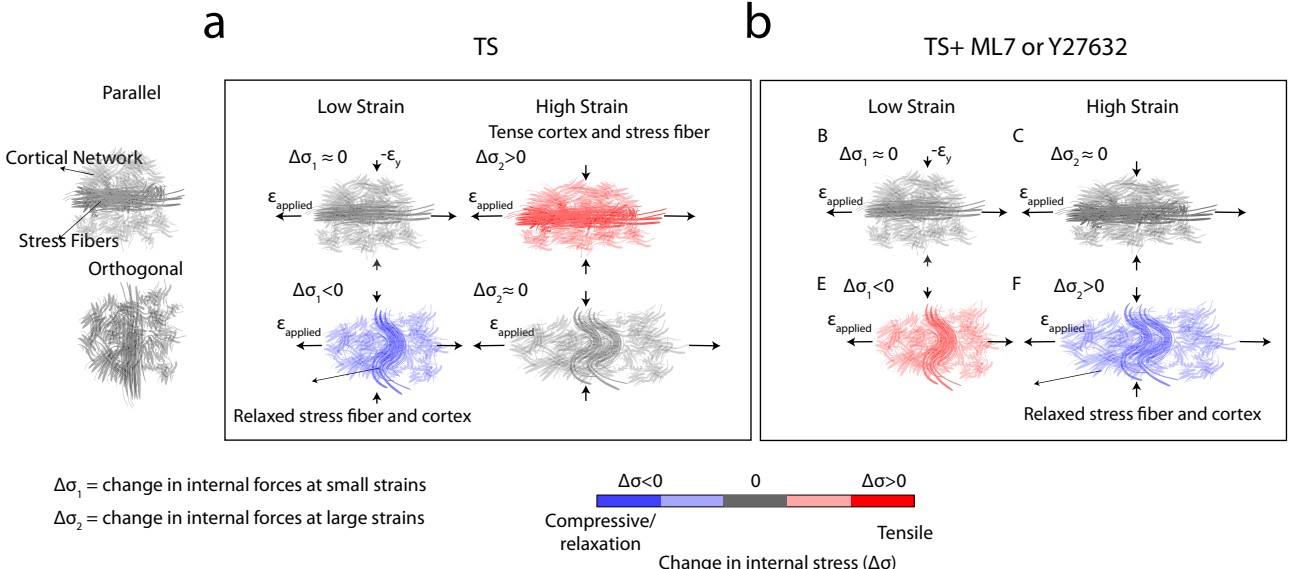

**Fig. 6 | The role of stress fibers and myosin activity in mechanical anisotropy of the cytoskeleton.** Integrated stress fiber and cortical network under (**a**) low and high applied strains parallel to the direction of stress fiber, (**b**) under ML7 or Y27632 treatment.

tension sensor's limitation to capture all intracellular tensions, leads to a dissimilar distribution of internal force.

Finally, we performed mechanical tests, in which strain is applied in the direction of SFs or orthogonal to it. Thus, SFs may either be compressed along their length (and stretched in the transverse direction) or extended along their length (and compressed in the transverse direction). From these tests, we see that the cell may exhibit a net increase or decrease in molecular tension, depending upon the direction of applied strain (Fig. 6a). The positive ΔFRET under 5-10% strains in orthogonal cells were compatible with mechanical relaxation in stretching cycles of cells where buckling events were identified under 10% compressive strain[89].

Parallel cells under ML7 or Y27632 treatments did not show a significant change in the molecular tensions which can be explained by a significant decrease in the elasticity of the stress fibers upon ML7 or Y27632 treatment[6,79]. This can be due to sliding of actin filaments past each other due to stretch upon inhibiting MLCK as observed in smooth muscle cells[80], or disassembly of contractile ventral stress fibers and transverse arcs[81]. Therefore, the effective stiffness of stress fibers in parallel strains might fall below the sensitivity threshold of the sensor, leading to the lack of significant measurable tension changes. This suggests that the mechanical tension in the parallel direction mainly originates from the myosin-generated forces. Orthogonal cells under ML7 or Y27632 treatments, however, exhibited molecular tension compared to non-treated cells, suggesting stretch-activated binding of myosin II[84], or regulation of myosin contraction through stretch-activated MLCK[82] (Fig. 6b, c) to resist the deformation. This suggests that the sensor not only detects myosin forces, but also captures applied forces to the cytoskeleton in the orthogonal direction. This indicates the force generation and resistance role of myosin in the observed anisotropy. As also previously shown in the context of cell stiffness, unlike Y27632, Blebbistatin induces softening in the cell cytoskeleton[78]. Blebbistatin-treated cells showed a relaxation response at high strains explainable by fluidization[90] due to direct inhibition of myosin ATPase and thus its binding to actin (Fig. 6b). At 20 μM Blebbistatin, the sensor exhibited little resistance to applied strains.

Thus, we suggest that when aligned with strain direction, SFs and cortex cooperate in load bearing. However, orthogonal to stretch, SFs relax while the cortex goes under molecular tension. These results indicate that while integrated structurally, the participation by distinct F-actin architectures in the bearing of the internal mechanical load is dynamic and varies with the state of external mechanical load. Our results are critical in understanding the mechanisms behind biological phenomena in which cells polarize and generate directional forces such as migration, reorientation, and polarization.

## Methods

### Construction and expression of tension sensor and control tension sensor in U2OS cells

The tension sensor (TS) and the control tension sensor (CTS) were made from the full-size human Filamin-A tension sensor using PCR-based oligonucleotide synthesis and confirmed by DNA sequencing. The elastic spring[34] between the EGFP and RFP was placed between the immunoglobulin domains 15 and 16. The elastic module is a 40-aa (GPGGA)$_8$ derived from the elastic spider silk protein flagelliform. The control tension sensor lacks domain Ig24 and Ig15 which has a ten amino acid linker (SGIDAALAAS) placed between the ABD (actin-binding domain) and GFP. Both constructs were cut out using the KpnI and NotI sites and ligated into mammalian expression vector pcDNA3 for transient expression.

Gateway recombination was used to generate the pLENTI plasmids of the TS and the CTS. Lentiviruses were made by co-transfecting HEK293T cells with packaging vectors psPAX2 (viral proteins Gag and Rev under the SV40 promoter; Addgene plasmid #12260, a gift from D. Trono, École Polytechnique Fédérale de Lausanne, Lausanne,

Switzerland) and pMD2.G (viral protein VSV-G expressed under the CMV promoter; Addgene plasmid #12259, a gift from D. Trono) together with the pLENTI constructs. Lentivirus particle-rich media was collected 48 h and 72 h after transfection and 0.45-μm filtered. U2OS HTB-96™ (ATTC) cell line was transduced by incubation with this supernatant (neat or diluted) and 8 μg/mL polybrene (Sigma Aldrich) for 18 h. Stable lines were established by puromycin selection. Cells were used after 2 weeks in culture. For transient transfection, 7.5 μl of Lipofectamine 3000 ® and 2.5 μg of the pcDNA3 plasmids were used according to the protocol, and cells were incubated for 24 h before imaging.

### Cell culture and FRET imaging and FRAPPA

Stably transfected U2OS HTB-96™ (from ATTC) cells (TS and CTS) are cultured at 37 °C under 95% air/ 5% CO$_2$ atmosphere in a culture medium consisting of Dulbecco's Modified Eagle Medium (DMEM) enriched with 10% FBS and 5% penicillin-streptomycin. As for additional load-free control, cells were lysed to calculate the FRET efficiency of the load-free sensor module. Detached and centrifuged cells were resuspended in a mixture of RIPA Lysis buffer (Thermo Scientific™) and protease inhibitor and incubated for 10 min at room temperature. In all, 30 μl drops of the solutions derived from TS and CTS cells were sandwiched between two coverslips.

The FRET imaging was conducted on a Yokogawa CSU-W1 spinning disk confocal imaging system (Leica Microsystems DMi8) equipped with an Andor Zyla 4.2 Megapixel sCMOS camera, and a Nikon TiE inverted confocal system with Yokogawa CSU-W1 spinning disk, equipped with an Andor iXon Ultra888 EMCCD camera using a 60x oil objective at 37 °C. Following the three-cube FRET imaging method[34,61], three images with 300 ms exposure time were taken of the samples in three separate channels. In the donor channel ($I_{DD}$), the excitation peak was at 488 nm, and the emission filter was at 527/55 nm. In the acceptor channel ($I_{AA}$), the excitation peak was at 560 nm, and the emission filter at 615/70 nm. Finally, the FRET channel ($I_{DA}$) was imaged with the excitation peak at 488 nm, and the emission filter at 615/70 nm. The laser intensities were set to 10% (5.0 mW) to get sufficient image and contrasts while avoiding photobleaching.

To calculate the tensor turnover time, FRAPPA was used with 405 nm laser on circular regions within the cells. The sensor recovery time ($\tau$) was calculated by measuring the RFP and GFP intensities over time, and best fitting to $a + b*e^{(-\frac{t}{\tau})}$.

### RFP dark and GFP dark mutants for inter/intradimeric FRET control

To evaluate the extent of FRET between individual tension sensors, we introduced mutations into the RFP and GFP fluorophores of the TS constructs to render them dark by QuickChange® site directed mutagenesis using primers listers in Supplementary Table 1. Subsequently, the plasmids were purified using miniprep and verified by Sanger Sequencing (Azenta Life Sciences). Finally, dark U2OS cells were co-transfected with TS-RFP dark and TS-GFP dark using Lipofectamine 3000 ® and the cells expressing comparable amounts of the TS-RFP dark and TS-GFP dark were chosen for imaging to evaluate the inter/intradimeric FRET.

### Construction of short- and long-length constructs for FRET efficiency controls

To calibrate the TS for FRET efficiency, we replaced the F40 elastic module (GPGGA)$_8$ in the original TS with either a short (GGSGGS)$_2$ linker – for high FRET or TRAF 2 domain of TRAF protein (TRAF) as a long linker for low FRET. Substitution of (GPGGA)$_8$ with (GGSGGS)$_2$ was performed with a Q5 site directed mutagenesis kit (New England BioLabs, Inc). To replace (GPGGA)$_8$ with TRAF2TRAF, TS without (GPGGA)$_8$ module and the TRAF2TRAF sequence from CTV (Addgene #27803) were PCR amplified and assembled with NEBuilder HiFi DNA

assembly master mix (New England BioLabs, Inc). To construct the freely diffusible FRET control with only GFP and RFP connected by spacers, GFP-(GSGGGS)$_2$-RFP and GFP-TRAF-RFP were amplified from the above two constructs and assembled into the pcDNA3 backbone using NEBuilder HiFi DNA assembly master mix. All constructs were confirmed by Sanger Sequencing (Azenta Life Sciences/ Keck Biotechnology Resource Laboratory). All primers used are listed in Supplementary Table 1.

### FRET analysis
FRET analysis was performed through pixel-by-pixel intensity-based calculation after spectral bleed-through and background subtraction corrections on the images taken according to the method mentioned in the FRET Imaging section. A linear bleed-through analysis was performed on EGFP-transfected or RFP transfected cells by taking images in the donor ($I_{DD}$), acceptor ($I_{AA}$) and FRET ($I_{DA}$) channels. The donor bleed-though ($d$) was calculated from the slope of pixel-wise intensity of FRET to donor channels from EGFP-expressing cells, averaged over ($n = 10$) cells (Supplementary Fig. 5). Similarly, the acceptor cross-excitation ($a$) was calculated from the slope of the pixel-wise intensity of FRET to donor channels from RFP-expressing cells (Supplementary Fig. 5). For our experiments and imaging setup, the bleed-through constants were $a \approx d \approx 0.045$. The corrected FRET intensity is then calculated using:

$$FRET_c = I_{DA} - a \times I_{AA} - d \times I_{DD} \qquad (2)$$

where, $FRET_c$ is the corrected FRET image. The FRET index is then calculated by normalizing the corrected FRET intensities to $I_{AA}$ (proportional to the amount of sensor):

$$FRET\ index = FRET_c / I_{AA} \qquad (3)$$

To measure the FRET efficiency, we used TS-(GSSGSS)$_2$ and TS-TRAF expressing cells to calculate the proportionality constant ratio $G$ between sensitized acceptor emission and quenched donor emission caused by FRET[61, 62]. The FRET efficiency is then calculated by:

$$FRET\ E = \frac{FRET_c / G}{I_{DD} + FRET_c / G} \qquad (4)$$

### Micropatterning of polyacrylamide gels and cell seeding
Quartz mask patterns (Applied Image, Rochester, NY) of circles, triangles, and squares were first cleaned with Kimwipes and water. Then the masks were wiped with pure acetone, ethanol, and anhydrous hexane, respectively to wipe out all the dirt and passivate the quartz glass to the polyacrylamide gels. 25 mm diameter glass coverslips were washed with soap for 30 min, followed by pure ethanol for 30 min. To make the coverslip surfaces reactive to polyacrylamide gels, they were treated with a combination of aminopropylsilane (Sigma Aldrich) and glutaraldehyde (Electron Microscopy Sciences). 2.8 kPa gels were used in the study with the ratios of polyacrylamide to bis-acrylamide 7.5%:0.1%. Finally, a solution with 0.05% w/v ammonium persulfate and 20 nM carboxylate-modified far-red beads of 40 nm diameter were made prior to polymerization. 15 μL drops of the gel and beads solution were placed on the areas of interest of the quartz masks and immediately sandwiched by the glass coverslips to polymerize for 30 min, while covered with a humid cap. Then the coverslips attached to the quartz glasses were flipped and exposed to preheated (for 4 min) deep UV light for 2 min (mirror side of the quartz glass facing the UV). It is noteworthy that UV exposure can somewhat increase the stiffness of the polyacrylamide gels[91]. The quartz masks were immediately placed in a bath of water for 15 min. The coverslips were peeled off carefully and sandwiched 100 μl drops of a mixture of EDC/NHS crosslinker

agents on Parafilm (850 μL DI water + 50 μL EDC (100 mg/mL) + 100 μL NHS (100 mg/mL)), covered with humid caps for 15 min. The coverslips were then washed and rinsed with DI water 3 times and sandwiched 100 μL drops of a well-mixed solution of bovine gelatin (0.2 mg/mL dissolved in 10 mM HEPES pH 7) as the adhesion agent for 1 hour at room temperature. The coverslips were then washed 3 times in cold HEPES 10 mM followed by 3 times in 1X PBS. The coverslips were immersed in 1X PBS and exposed to UV light for 15 min to sterile and stored in the cold room. Finally, 500 μL drops of U2OS FRET TS stable cell with a density of ≈3000 cells/μL were seeded on the coverslips. The cells were then imaged 3-6 hours after seeding. Cells that did not completely spread to the micropatterned area or failed to exhibit responsive behavior to drug treatments based on their tractions were excluded from the micropatterning experiments.

### Immunofluorescence staining and colocalization analysis
Cells were fixed with a solution of 4% paraformaldehyde in 1X PBS for 15 min and permeabilized with 0.2% Triton X-100 for 20 min. The cells were then blocked with 2% bovine serum albumin (BSA) in 1X PBS (PBS-2%BSA) at room temperature for 1 h. Phalloidin staining was performed with Alexa Fluor 405 Phalloidin (Life Technologies; 1:1000) diluted in PBS with 2% BSA for 2 hours at room temperature. Paxillin staining was performed with anti-Paxillin (Abcam ab32084, Knockout validated) and secondary Alexa Fluor 647 (Abcam ab150075). For 50 μM SMIFH2 treatment, cell were consequently incubated for 1 h. Images were taken with a 60x oil immersion objective.

Colocalization (spatial correlation) analysis is done using a two-dimensional Pearson's correlation coefficient between two channels using the following equation:

$$C_{A/B} = \frac{\sum_m \sum_n (A_{mn} - \bar{A})(B_{mn} - \bar{B})}{\sqrt{\left(\sum_m \sum_n (A_{mn} - \bar{A})\right)^2 \left(\sum_m \sum_n (B_{mn} - \bar{B})\right)^2}} \qquad (5)$$

Where $m$ and $n$ are the image dimensions, $A$ and $B$ are the image intensity of either of the channels to be correlated, $\bar{A}$ and $\bar{B}$ are the mean value of intensities over each channel. Positive values indicate a direct correlation between channels and negative values show an anti-correlation between two channels.

### Traction force microscopy
The tractions induced by cells on the substrate were measured using the traction force microscopy method[92]. Deformed substrate images of far-red fluorescent (excitation 560 nm, 200 nm diameter) beads embedded in the PAA gels were obtained using a 60x oil-immersion objective.

10 nM Calyculin-A (abcam ab141784) and 10 μM Blebbistatin (B05600) were used to regulate intracellular forces. The zero-traction image (reference) was obtained at the end of each experiment by adding trypsin to the cells for 1 hour to detach from the substrate. The images were aligned and registered to correct for the drift in imaging, and a Particle Image Velocimetry analysis (mPIV[93]) was performed between each image and the reference image to obtain the deformation vectors, on grids of size 35 μm. The traction vectors were then calculated using a MATLAB® code written by Ulrich Schwarz[92].

### Alignment order and stress fiber segmentation
To quantify the alignment of the internal actin cytoskeleton, the alignment order was calculated in MATLAB®. After applying a Gaussian filter on raw images, the image is binned to small windows of 16 pixels, with 50% adjacent overlaps. A 2D Fast Fourier Transform (FFT) was then applied to all the windows, and the axis of the least second moments was calculated to determine the angle of the orientation in each window. Next, the local and global degrees of alignment were

calculated according to:

$$\langle \phi \rangle = 2\left\langle \cos^2 \delta\theta - \frac{1}{2}\right\rangle, \delta\theta = \theta_i - \theta_j, \qquad (6)$$

at a window centered to the point $x_0$ and $y_0$, where $\delta\theta$ is the angle difference of adjacent vectors within a window size of r, i.e., $(i,j) = ([x_0 - r/2, x_0 + r/2], [y_0 - r/2, y_0 + r/2])$. The windows at the boundary of the cell were similarly analyzed, instead by ignoring the NaN values outside the cell area where there is no alignment field. Thus, $\langle \varphi \rangle$ was calculated by taking the mean of $\langle \varphi \rangle$ over the entire area of the cell. We used $r = 5\mu m$ and $r = 10\mu m$ for the mean local and global order within the cell cytoskeleton.

To detect, segment, and mask the cortical actin and stress fibers, we used Fsegment[88], a MATLAB® written software (Supplementary Fig. 14). After converting the images to .BMP format, we found the default parameters in the detection algorithm working well for with our datasets. The software outputs binary masks of segmented stress fibers, and we inverse the masks to detect the cortical actin. Finally, these masks were used to measure the FRET E at the regions of interest.

### Cell stretching experiments
Cell stretching experiments were conducted using Cytostretcher from Curi Bio, Inc. Nano grooved (parallel and orthogonal) PDMS chambers were coated with 0.2 mg/mL gelatin for 24 hrs incubation at 4 °C. The chambers were then seeded with TS and CTS U2OS cells with 4 hour long attachment time window. The chambers were then mounted and clamped on the Cytostretcher. The stretch/relaxation protocol was set to 45 seconds of stretching and 20 seconds of relaxations, with a ramping time of 5 seconds using the Curi Bio software. The orthogonal grooved, vertical grooved, and smooth PDMS chambers were stretched for 1, 2, 3, and 4 mm. Time-lapse movies of stretched and relaxed states were taken with ×60 magnification objective and stored and analyzed through the same method mentioned below.

In control experiments, 67 μM ML7 (www.sigmaaldrich.com/US/en/product/sigma/i2764), 10 μM Y27632 (www.stemcell.com/products/y-27632.html), 200 μM CK666 (www.sigmaaldrich.com/US/en/product/sigma/sml0006), 10 μM and 20 μM Blebbistatin (www.sigmaaldrich.com/US/en/product/sigma/b0560) treatment for 30 min before the experiment was used after 4 hours of attachment. Under 20 μM Blebbistatin at 20% strain, cells started to detach from the substrate which was excluded from the data.

### Reporting summary
Further information on research design is available in the Nature Portfolio Reporting Summary linked to this article.

## Data availability
Raw data supporting the finding of this manuscript are available from the corresponding author upon request because of the large size of the FRET image data. The data generated in this study are provided in the Supplementary Information and Source Data file. Source data are provided with this paper.

## Code availability
Code supporting the findings of this manuscript is available from the corresponding authors upon request. A Reporting Summary for this Article is available.

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

## Acknowledgements

We acknowledge funding ARO MURI W911NF-14-1-0403 to M.M., S.A., and M.A.S. We also acknowledge the National Institutes of Health (NIH) R01 1R01GM126256, the National Institutes of Health (NIH) U54 CA209992, and the Human Frontiers Science Program (HFSP) grant number RGY0073/2018 to M.M. Any opinions, findings, conclusions, or recommendations expressed in this material are those of the authors(s) and do not necessarily reflect the views of the DoD, NIH, or HFSP. We would like to thank Dr. Corey O'Hern and Dr. Amir Pahlavan for constructive discussions. We also thank Dr. Daniel V. Iwamoto, and Dr. Abhishek Kumar for helpful discussions on the tension sensor assembly and Dr. Archer Hamidzadeh for helping with setting up the FRET imaging. We thank the Yale West Campus Imaging Core for the support and assistance in this work.

## Author contributions

M.M., S.A., C.M. and designed and conceived the experimental work. M.M., C.M., M.A.S., and D.C. designed and made the tension sensor. C.M. and C.C. developed the lentivirally-transduced cell lines. C.C. S.A. and C.M. developed the dark mutant controls. X.S. and S.A. developed the FRET efficiency constructs. S.A. and C.M. conducted the experiments. S.A. analyzed the experimental data. M.M. and S.A. drafted the paper. M.M., S.A., C.M., M.A.S., and D.A.C. edited the paper.

## Competing interests

The authors declare no competing interests.
