## [Peer Review File · Nature Communications]

Intracellular Tension Sensor Reveals Mechanical Anisotropy of the Actin CytoskeletonREVIEWER COMMENTS

Reviewer #1 (Remarks to the Author):

Amiri et al. developed a new tension sensor based on filamin A. The FRET efficiency changed according to the stress applied to the cells as anticipated. The effect of calyculin A and blebbistatin also agrees with the mode of action of this sensor. By using cells plated on PDMS substrate, the effect of shape and tension was extensively examined to conclude that the mechanical response is anisotropic at low strain and more isotropic at high strain. Overall this work has been well done in an organized way and adds new insight into the biophysics of actomyosin contraction. To the best of this reviewer's knowledge, the filamin-A sensor has not been published. Since a major claim of this work is the development of the new sensor, detailed information on this sensor and the difference from the previous ones is required in this paper. Other than this, this reviewer has only minor comments to improve this paper.

Re: tension sensor

1. What is the difference between the Talin- and Filamin A-based tension sensors? Why did the author need to develop a new tension sensor?
2. The Talin-based sensor localizes at stress fiber by means of actin-binding domain and vinculin-binding sites. Is the actin binding site of Filamin-A sufficient to bring the sensor to the stress fiber? To answer this question, images of the control tension sensor is required in Fig. 1C,.

Comments:

1. Line 51: What is the GEF pathway? The reference cited here may not match.
2. Line 91: SMIFH2 is not described in M&M.
3. Line 113: This sentence does not match the content of Fig. S3.
4. Line 119: The first and second equations appear to be incorrect.
5. Line 174: This sentence does not match Fig. S5.
6. Supplementary Fig. 4 is not referred to in the text.
7. Line 230: Martin Schwartz is listed as the co-author, therefore he should not be acknowledged in the M&M section.

8. Line 247: DMI8 is the brand name for Leica's inverted microscope and does not specify the spinning disk confocal unit. A detailed description will be required.

9. Supplementary Fig. 2: If the color of each image reflects the wavelength of fluorescence, panels A and D should be shown in red. Honestly, this reviewer does not understand the intention to show these panels.

Reviewer #2 (Remarks to the Author):

The manuscript "Intracellular Mechanical Stress Measurements Reveal Strain-dependent Anisotropy of the Cell Cytoskeleton" describes a very nice approach to combining traction force microscopy with molecular tension sensors and intracellular stress inference to improve the understanding of force and stress transmission inside cells. The paper is well written, but shows some surprising mistakes, like a forgotten reference to a figure. The subject is very important, as we currently lack any clear information about intracellular force distribution, hence the scope of the paper is excellent. Here the authors make use of a new tension sensor that exploits the actin crosslinker filamin to determine the force transmission within cells. While I appreciate a lot the effort, I am mainly worried about the readout of the FRET sensor. Since the first introduction of the sensors, it became clear that new sensors must be very carefully tested and crosschecked with a series of control experiments. Unfortunately, most of these controls have been omitted here, which is especially surprising as probably the best expert on these sensors is a co-author of the paper. Please find below a list of main and minor issues.

Main issues:

- FRET Sensor. The control is not putting the FRET pair at the end of the molecule as typically done. In this situation, we do not know about the possible interaction of domain 16 with any intracellular structure, thus making the control sensor questionable. Can the authors be sure that this domain does not interact with other structures in the cell?

Furthermore remains the problem of intermolecular FRET. The data shows a high decoration of the sensor to the stressfibers. It would be important to understand to which extent neighbor sensors interact. This can be nicely tested by using a pair of control constructs (see: DOI: 10.1007/s12195-014-0368-1). Another important check would be to see if the

sensors are actually functional. Although the authors nicely show the colocalization with the stress fibers it is not clear if they can really fulfill the same role as a normal filamin molecule. Ideally would be here to perform some KO and rescue experiments of phenotypes. I am aware that these control experiments are tedious, but I am convinced that they are important given the reported difficulties of such sensors when not checked carefully.

- Although the intracellular stress inference is a very interesting approach, the results suggest that it is not relevant as it does not take the mechanical configuration of the cell into account. I suggest pointing this out even more pronounced than already done.

- I got confused with the way the order parameter was calculated. Unfortunately, although the explanation seems to have been due to be included in the Materials and Methods (line 298), it was not explained there. Does the local parameter refer to $5\mu\text{m}$ around the center of the cell? Or was the center of the $5\mu\text{m}$ radius region moved over the cells? In this case, what happens when the center of this point is at the cell edge? The same accounts for the global order parameter.

- The authors simply assume that all regions without stress fibers are dominated by cortical actin. What about intracellular actin structures? Or asked differently, is it valid to only separate the actin signal into stress fibers and cortex without including intracellular networks?

- Do the normalized values of sensor enrichment in Figure 1E,F make sense? It will light up even at little TS in regions with little actin, so that unbound TS background would be mistaken as highly enriched regions. Why is this shown? Is not a correlation better? What does the I on the y-axis of Fig 1.G and H stand for? The short reference to this in the caption is a bit confusing.

- It seems that the cell shown in video 2 contracts at timepoint 1:44, hence after the addition of calyculin (which makes sense). Should this not lead to an artificial increase in average traction as the area is decreasing? Effectively, it could lead to a situation where the TFM does not really increase in absolute values. To which extent does this influence the analysis?

- In Figure SI5 we see the quantification of Delta RFP, but it remains unclear how this was done or what it was used for. I expect that the total intensity of RFP is constant, so why should it change?

- The stop of FRET signal at $20\mu\text{m}$ Blebbistatin for any applied external strain is worrying me.

This suggests that the FRET sensor depends on Myosin. The way the authors introduce the sensor suggests that they should bind to neighboring filaments, and these should still be stretched when the cell is stretched. Potentially, the sensor depends directly on myosin, and then we would measure not the force between actin but between actin and myosin. Here the functional test of the sensor as suggested above as control experiments would be important.

- The authors mention that the sensors partially unbind in parallel stretched cells referring to SI Fig 5. But how is this unbinding measured?

- End of the results section, the authors use the found correlation between the SF length and the FRET signal. It is not clear how the total length is measured. Moreover, the authors take a strong statement from this correlation by stating that the amount of SF regulates the changes in stress response to strain. A simple correlation is not enough to infer causality. Please provide further data or remove this overstatement.

- Why do the authors not refer to figure 6 somewhere in the text?

- The authors use a UV illumination of the PAA to activate the patterns. Unfortunately, the authors do not provide information about the light dose and wavelength. In my experience, deep UV illumination of PAA to pattern the surface leads to changes in elasticity because the deep UV breaks chemical links. Did the authors quantify the stiffness of the gel at the patterns, e.g. by AFM?

- Overall, the authors report many average values without showing any error values. It would be important to also give an estimate of the validity of the mean value of TFM and FRET, for example by also showing the standard deviation.

- Figure 5H, seems to be inconsistent with Figure 4G where the FRET values go to numbers exceeding 0.2.

Minor:

- Line 103. Are the authors referring to Fig. 2D? There is no dashed line in 2E

- Line 114: Typo, "compared fluorescence the intensity" -> "compared the fluorescence intensity"

- SI Fig 4B, please define how the correlations are calculated. The definition of the order parameter remains unclear.

- Line 144: Typo "...to their direction of cell polarization" -> "...to the direction of cell

polarization”

- Typo: Laica -> Leica

Reviewer #3 (Remarks to the Author):

SUMMARY

The authors created a novel FRET-based tension sensor based on a shortened version of filamin. The authors then attempt to use the sensor to learn about the role of anisotropy in intracellular mechanics. The comparison of traction forces and intracellular stresses is an important, but understudied, aspect of cell mechanics. However, the manuscript uses imprecise terminology when comparing molecular and macroscopic mechanical variables, lacks many controls for FRET-based tension sensors, and some of the calculations involving the interpretation of FRET values have errors (e.g. incorrect formulae). These major errors must be remedied before the manuscript can be fully evaluated.

MAJOR CONCERN

1. The authors switch between using mechanical terms that are best defined at large length scales in two or three dimensions where continuum mechanics approaches (e.g., stress, strain...) are appropriate and a molecular description that is appropriate for describing the FRET-based tension sensor (tension, compression...). This needs to be resolved to prevent many potential issues in interpretation of the results. In this review, I will place the word “molecular” before mechanical variables when I am referencing the sensor. I suggest the authors first be more precise in the use of mechanical terminology across length scales and adopted a similar convention of their choice.

2. It has been reported that the molecular tension sensors can report molecular compression (i.e. a negative tension in a one dimensional sense) (see doi: 10.1038/nature13535 and 10.1007/s12195-015-0404-9). Thus, it is critical to establish a value of FRET corresponding to the unloaded state. FRET values less than this value are consistent with the sensors experiencing molecular tension (on average, within an imaging voxel). Higher values are constituent with molecular compression of the sensor and/or the onset of intermolecular FRET (more on this below). Thus, it is very problematic that

unnormalized FRET index values shown in Fig 2J are so low for the presumably unloaded CTS. Specifically, the presumably unloaded CTS values have FRET indices of approximately 0.15. The FRET values for TS at low traction forces are as high as 0.6. This would suggest that either CTS is not functioning as expected or significant molecular compression in the sensor and/or the presence of substantial intermolecular FRET in the TS. Currently, these discrepancies render all FRET data in the manuscript uninterpretable, and this issue must be addressed before publication.

3. One possibility for the discrepancy in FRET values for the TS and CTS is that the TS is a dimer and undergoing intermolecular FRET. This would drastically affect the interpretation of the data. As has been specified in multiple reviews on the creation and use of FRET-based tension sensors, the authors should perform intermolecular FRET controls involving dark mutants of the various FPs (e.g., 10.1007/s12195-014-0368-1). Notably, this approach was pioneered in the Schwartz Lab (10.1016/j.cub.2013.04.049). In the opinion of this reviewer, this control must be performed for the manuscript to be suitable for publication.

4. Furthermore, the authors have chosen to represent the majority of their FRET data as “change in FRET”. This prevents discerning of the absolute values of the FRET index. This leads to problems in the conceptualization of the results, as in the context of FRET-based tension sensors “molecular compression” has been used to describe sensors are pushed closer together than their unloaded resting state. The context of “compression” in this manuscript is not immediately clear to this reviewer. I think, but I am no 100% sure, that the authors are using “compression” to mean “increase in FRET” from a loaded baseline. Thus, “compression” could refer to reduction of tension (e.g. from high to low molecular tension, instead of the onset of molecular compression). Furthermore, the relationship between FRET index and molecular tension is not linear and is unknown for molecular compression (see <https://doi.org/10.7554/eLife.33927>). Thus, it is unclear what the change in FRET Index metric actually represents (see below for more regarding the calculations in the supplement). The terminology issue, as well as an inability to discern the absolute value of the FRET indices, should be fixed to enable interpretation of the data in the manuscript.

5. The idea of comparing the traction forces, internal “stresses” determined from inference methods, and the FRET signal is interesting. But, it is unclear this analysis can be interpreted for multiple reasons. To begin, I will discuss the mechanical issues. It is completely unclear if the sensors report all internal stresses in the cell. First, there are a plethora of endogenous

actin cross-linking proteins. These could all be bearing portions of the intracellular stresses not detected by molecular tension sensor. Secondly, the sensor probably estimates forces between F-actin filaments, and not forces transmitted along filaments, which might be expected to dominate in stress fibers. Thirdly, as shown in Figure 1, the authors demonstrate that the sensor does not equally label all actin structures. Thus, components contributing to external stress field could be missed by the sensor. Next, I will discuss it from the sensor side. There are many issues due to the dynamic binding of the sensor. As shown in the supplemental material, the binding time of the sensors is quite short. Thus, sensors could be bound to no filaments (i.e. soluble) or bound to a single filament. The relationship of sensors in these states to stress measurements is unclear. Also, there are issues with the sensitivity of the tension sensing modules. The authors used FRET Index for this comparison. FRET Index is not simply related to FRET Efficiency (10.1529/biophysj.106.088773). Furthermore, FRET Efficiency is non-linearly related to the underlying molecular tension (10.1038/nature09198, <https://doi.org/10.7554/eLife.33927>). It is unclear how not accounting for these complex relationships will affect the comparison between FRET signals, intracellular stresses, and traction forces. Additionally, the proposed analysis requires that the sensors accurately report the actual molecular tension that it reports. This sensitivity of the sensor is like in 1-6 pN range, and molecular forces outside that range cannot be measured. Furthermore, no sensor has been calibrated for molecular compression. Thus, if there are sensors in molecular compression, how their FRET signal contributes to estimates of mechanical stress is completely unclear. Additionally, any contribution from intermolecular FRET would bias these comparisons as well. For all these reasons, this reviewer suggests that figure 3 be re-done with a more correct analysis (see below, potentially involving FRET efficiency if concerns regarding intermolecular FRET and molecular compression can be addressed) and then added as supplemental figure. I expect that the inferred internal stresses will not match the tension sensor signal, but, due to the issues listed here, it is unclear how to interpret that result. However, it clearly cannot be interpreted as due to a lack of consideration of anisotropy alone. Furthermore, these inference methods also ignore all dissipation of mechanical energy of the system as well as local variation in the mechanical properties of the system.

6. There are several errors in the Supplemental Note 2. The molecular tension sensors do not act like linear springs in the cell domain, and the referenced citation does not show that.

The extensible domains of the tension sensors act like unstructured polypeptides that can be well described using models of worm-like chains from polymer physics. See <https://doi.org/10.7554/eLife.33927> for details. Secondly, the equation relating FRET index and FRET Efficiency is incorrect. These variables are not directly proportional, and the cited reference does not state that they are. The relationship between FRET efficiency, the various imaging channels, and the correction factor G can be found in Chen, *Biophysical Journal*, 2006 (<https://doi.org/10.1529/biophysj.106.088773>). It is not clear to this reviewer what if any data these errors effect, but they should be corrected and effected analyses re-done. The errors do seem to affect how the authors are interpreting their data. For instance, I believe these calculations are the basis for interpreting the change in FRET in terms of mechanical variables. Thus, the interpretation of the data and discussion should be re-considered once the errors are corrected.

7. If the authors have used “three-cube” FRET imaging as stated in the methods, then they should be able to readily calculate FRET efficiency. This is described in <https://doi.org/10.1529/biophysj.106.088773>, and has been used in the context of FRET-based tension sensors (<https://doi.org/10.7554/eLife.33927>, <https://doi.org/10.1002/cyto.a.23688>). These FRET efficiencies can be converted to the molecular tension experienced by the sensor if the tension sensing module has been suitably calibrated.

8. The authors first state the slope (m) shown in Figure 5, can interpreted as indicating the degree of load bearing by the various structures. I agree with this interpretation. However, they offer an alternative interpretation that says that the data can be used to infer the stiffness of the structures. This later interpretation seems to assume that the two structures are subject to same mechanical input. However, this is not necessarily true as the stretching forces are transmitted to the various cytoskeletal elements through adhesion receptors, and coupling between the two structures and the adhesions may not be uniform. It is opinion of this reviewer that either experiments directly probing the stiffness of the various actin structures be added to the manuscript or this alternative interpretation be removed from the paper. Specifically, without additional data, Figs. 5 J and K need to be re-stated in just terms of differential load-bearing.

9. I believe the main point of Fig.6 is that there is an unexpected increase in tension in the orthogonal direction for large strains. This effect does not appear to occur in blebbistatin-

treated cells (Fig.4I). Furthermore, the response is reversed if ML-7 is used to inhibit MLCK, and the expected increase in FRET is observed. I believe this suggests a counter hypothesis that at high strains the cells activate myosin activity (likely through MLCK) to globally increase molecular tension. This alternatively hypothesis should be excluded or at least discussed. One suggested (but not required) experiment would be to address the role of stretch activated calcium channels in this system, which could be activated at large deformations. MLCK is associated with Calcium-based regulation of contractility, so this connection could explain the primary observation. Regardless of the additional experiments, this reviewer suggests that the inhibitor data be included in the summary Figure 6 and the statement of the primary results of this manuscript.

MINOR CONCERNS

1. In Fig. 2C it is not immediately clear what is shown, as “Control” could be in reference to the use of CTS, the lack of an inhibitor treatment, or both.
2. The text states that “We verified that the sensor remains bound in orthogonal cells, however, it partially unbinds in parallel cells (Supplementary Figure 5).” The opposite result seems to be shown in that Figure. Also, it seems that the error bars are much larger in C than in B. This could be important as the size of the effect in C is often the same as in B. The authors should elaborate on the origin of the different variances as well.
3. The authors should specify which type of blebbistatin that was used for live cell imaging. Classical blebbistatin can be toxic in live cell assays and can lead to substantial background signal in FRET experiments.

RECOMMENDATIONS

4. There are many experimental conditions but relatively few pictures of cells. In the context of Fig. 5, it would be nice to have more pictures of the cells to create a clear picture for the reader. For instance, do the cells treated with blebbistatin or ML-7 have altered stress fibers?
5. The paper would be strengthened with the inclusion of experiments addressing the role of RhoA-ROCK mediated myosin activation, as in addition to MLCK, this is the other major pathway of regulating myosin activity. In particular, experiments with a ROCK inhibitor would be interesting.

6. In the opinion of this reviewer, the most interesting data in the manuscript is that fact that there are multiple relationships between the FRET signal of the sensor and traction force, as shown in Fig. 2J. It would be good for the field to discuss this observation more.

REVIEWER COMMENTS

Reviewer #1 (Remarks to the Author):

Amiri et al. developed a new tension sensor based on filamin A. The FRET efficiency changed according to the stress applied to the cells as anticipated. The effect of calyculin A and blebbistatin also agrees with the mode of action of this sensor. By using cells plated on PDMS substrate, the effect of shape and tension was extensively examined to conclude that the mechanical response is anisotropic at low strain and more isotropic at high strain. Overall, this work has been well done in an organized way and adds new insight into the biophysics of actomyosin contraction. To the best of this reviewer's knowledge, the filamin-A sensor has not been published. Since a major claim of this work is the development of the new sensor, detailed information on this sensor and the difference from the previous ones is required in this paper. Other than this, this reviewer has only minor comments to improve this paper.

We would like to thank the reviewer for identifying our work as insightful and novel. Below, we have tried our best to address all the reviewer's comments, which we think has significantly improved the paper.

We should also note that we have incorporated Reviewer 3's comments which required us to convert the FRET measurements to FRET efficiency, as a more reliable metric than the FRET index. This required us to add additional controls and experiments, which strengthened our results. We have added all the relevant information to Supplementary Figures and updated the Material and Methods section. After the new analysis, **our salient results have remained consistent**. The results with the FRET index have thus been moved into Supplemental Figures.

Re: tension sensor

1. What is the difference between the Talin- and Filamin A-based tension sensors? Why did the author need to develop a new tension sensor?

Talin-based tension sensors are used to measure forces between F-actin and integrins at the basal surface. The filamin A-based tension sensor measures forces inside the actin cytoskeleton, essentially between nearby actin filaments. The two sensors thus measure forces in two distinct domains.

“To date, the measurement of cell-generated mechanical forces is principally limited to those exerted against the extracellular matrix (ECM), within focal assemblies^{22,23}, receptors, or receptor-bound linkers²²⁻²⁴”

2. The Talin-based sensor localizes at stress fiber by means of actin-binding domain and vinculin-binding sites. Is the actin binding site of Filamin-A sufficient to bring the sensor to the stress fiber? To answer this question, images of the control tension sensor is required in Fig. 1C,.

We thank the reviewer for this insightful comment. We have now added images of the CTS cells to Figure 1D as requested, and the enrichment analysis in Fig 1H includes the results for the CTS cells as well (the square and triangle shapes are moved to Supplemental Figure 3). Although these images show somewhat higher cytosolic CTS, compared to TS, the colocalization analysis (Fig 1I) shows that there is still a high and significant correlation between the sensor and actin. The increased cytosolic localization for the control sensor is very likely due to the absence of the dimerization domain. We also show the freely seeded (unpatterned) TS and CTS cells in which the cytosolic sensor is visible (Supplementary Figure 2).

Comments:

1. Line 51: What is the GEF pathway? The reference cited here may not match.

Thanks for the comment. We have clarified that sentence, now it reads:

“Yet, anisotropies and nonlinearities are thought to play major roles in cellular mechanobiology, including in the strain-dependent gene expression in stem cells^{31–34}, regulation of YAP mechano-transduction³⁵, and GEF binding^{36,37} that activates Rho GTPases to control cytoskeletal dynamics”

2. Line 91: SMIFH2 is not described in M&M.3. Line 113: This sentence does not match the content of Fig. S3.

Added the SMIFH2 control to the Material and Methods “**Immunofluorescence staining and colocalization analysis**”.

Line 113 in the original submission referred to the different slopes and thus non-unique FRET index for a single traction stress (old Figure 2J). Based on suggestions from Reviewer 3, we have updated our results in terms of FRET efficiency. We have also swapped the FRET efficiency and $\langle |T| \rangle$ in Fig 2J. We show that FRET efficiency is less sensitive to the expression levels than the FRET index (Supplementary Figure 9), thus we switched the results to FRET efficiency. In the updated plot with FRET efficiency, we still see a significantly different slope between the calyculin vs blebbistatin treated cells.

4. Line 119: The first and second equations appear to be incorrect.

We understand the objection is about the sign of T_x and T_y . From the TFM method, we typically get inward contractile traction stress vectors which are applied from cell to substrate. Therefore, we added a negative sign to account for the reaction forces applied from the substrate to the cell. Now it reads:

“Where T_x and T_y are the x and y components of the traction stress vectors applied to the substrate by cells”

5. Line 174: This sentence does not match Fig. S5.

We have revised our explanation in the updated Supplementary Figure 12, which includes the relative changes in total intensity of the sensor (ΔRFP) and the density ($\Delta(RFP/A)$) of the sensor within the cells before and after stretching. Now it reads:

“Changes in the FRET E may also be due to the unbinding/binding of the sensor or appearance of cytosolic sensors in the focal plane. The quantification of the binding/unbinding of the sensor was performed by measuring total amount of the sensor ΔRFP and the density of the sensor $\Delta RFP/A$, where A is the cell area. With $\Delta RFP \approx -0.05$, we verified that in orthogonal cells, despite a basal level of sensor unbinding, the sensor remains bound during stretching (Supplementary Figure 12). Decreasing $\Delta RFP/A$ (≈ -0.10 , ≈ -0.16 , ≈ -0.21 , ≈ -0.31 for at 5%, 10%, 15% and 20% strain), further verified that the density of the sensor decreases with strain, as the area increases with strain. However, in parallel cells there is an increase in the density ($\Delta RFP/A \approx 0.2$) and the total amount of the sensor ($\Delta RFP \approx 0.2$), suggesting a deformation-induced revealing

of binding sites on actin or integrins^{66,67} (Supplementary Figure 12). As this effect is not observed in the orthogonal direction, we suggest the elongation of actin-rich stress fibers which also colocalize with integrin sites reveals the majority of binding sites for the new sensors to attach.”

6. Supplementary Fig. 4 is not referred to in the text. In the original submission, line 135 referred to Supplementary Figure 4. It is now renumbered as Supplementary Figure 11.

7. Line 230: Martin Schwartz is listed as the co-author, therefore he should not be acknowledged in the M&M section. Corrected.

8. Line 247: DMI8 is the brand name for Leica’s inverted microscope and does not specify the spinning disk confocal unit. A detailed description will be required. We have added this information.

9. Supplementary Fig. 2: If the color of each image reflects the wavelength of fluorescence, panels A and D should be shown in red. Honestly, this reviewer does not understand the intention to show these panels.

We show the color based on the emission wavelength, which is green for GFP and FRET, and red for RFP. These panels show the EGFP and RFP transfected cells, imaged by the FRET Imaging protocol (Material and Methods). This is to calculate the undesired bleed-through of the red and green fluorophores into the FRET channel and correct the FRET analysis for this effect.

Reviewer #2 (Remarks to the Author):

The manuscript “Intracellular Mechanical Stress Measurements Reveal Strain-dependent Anisotropy of the Cell Cytoskeleton” describes a very nice approach to combining traction force microscopy with molecular tension sensors and intracellular stress inference to improve the understanding of force and stress transmission inside cells. The paper is well written, but shows some surprising mistakes, like a forgotten reference to a figure. The subject is very important, as we currently lack any clear information about intracellular force distribution, hence the scope of the paper is excellent. Here the authors make use of a new tension sensor that exploits the actin crosslinker filamin to determine the force transmission within cells. While I appreciate a lot the effort, I am mainly worried about the readout of the FRET sensor. Since the first introduction of the sensors, it became clear that new sensors must be very carefully tested and crosschecked with a series of control experiments. Unfortunately, most of these controls have been omitted here, which is especially surprising as probably the best expert on these sensors is a co-author of the paper.

We thank the reviewer for emphasizing the importance of the topic and highlighting the strengths of the work. We have done our best to address all these comments and feel that the manuscript is significantly improved.

We also note that we have incorporated Reviewer 3’s comments, which required converting FRET Index to FRET efficiency as a more reliable metric. This required additional controls and experiments, which strengthened our results. We have added all the relevant information to the Supplementary Figures and updated the Material and Methods. After the new analysis, **our salient results have remained consistent**. Older results with the FRET index have been moved to the supplement.

Please find below a list of main and minor issues.

Main issues:

- FRET Sensor. The control is not putting the FRET pair at the end of the molecule as typically done. In this situation, we do not know about the possible interaction of domain 16 with any intracellular structure, thus making the control sensor questionable. Can the authors be sure that this domain does not interact with other structures in the cell?

We thank the reviewer for this comment. The reason why we put the FRET pair between the Ig15 and Ig16 is the H1 domain, which is a flexible domain. We did not want to change the dynamics of the protein. Moreover, we could not put the tension sensor module at the end of the molecule because this could affect domain Ig24, which is the dimerization domain. Without dimerization, the construct does not measure molecular force. It is known that Ig16 does not bind actin, it binds transcription factors such as androgen receptor (AR) and FOXC1, ECSM2, an exclusively endothelial-specific surface protein, and FLNA-interacting protein (FILIP) (doi:10.1016/j.tcb.2009.12.001).

To address this, we show that the control sensor has the same FRET E as the sensor module alone, and thus does not appear to interact with any cellular structures that would result in force application. Moreover, we showed the CTS construct does not significantly report any changes in FRET E, therefore we suggest that domain 16 does not appear to interact with any cellular structure that result in force application. Furthermore, both the TS and CTS have the domain 16, and as we have shown the TS reasonably responds to actomyosin force. Therefore, we believe the potential interactions of domain 16 should not be a problem

in our force measurements. We have added a sentence that acknowledges this possibility in the main text. Now it reads:

“We also note that Ig16 has potential binding interactions with certain intracellular components⁵³, which may invalidate the assumption that the CTS functions as a zero-force control. To address this, we conducted FRET E measurement on lysed cells, in which the force sensing module is diffuse, and thus experiences zero force. (Material and Methods). We found that lysed TS and CTS cells exhibited insignificant differences from intact CTS cells (Supplementary Figure 8). These results confirmed that the CTS construct is not affected by myosin-dependent contractility and the potential interaction of Ig16 with intracellular structures⁵³ is negligible in our analysis.”

Furthermore, remains the problem of intermolecular FRET. The data shows a high decoration of the sensor to the stressfibers. It would be important to understand to which extent neighbor sensors interact. This can be nicely tested by using a pair of control constructs (see: DOI: 10.1007/s12195-014-0368-1). Another important check would be to see if the sensors are actually functional. Although the authors nicely show the colocalization with the stress fibers it is not clear if they can really fulfill the same role as a normal filamin molecule. Ideally would be here to perform some KO and rescue experiments of phenotypes. I am aware that these control experiments are tedious, but I am convinced that they are important given the reported difficulties of such sensors when not checked carefully.

We thank the reviewer for this insightful comment. We have performed a new control experiment to measure intermolecular FRET. Briefly, following the method mentioned in the suggested paper, we made two new constructs in which either RFP or GFP was mutated to eliminate fluorescence. Co-expressing these constructs in dark U2OS cells using transient Lipofectamin transfection, we verify that the intermolecular FRET does not play a role in our analysis. We have added these results to Supplementary Figure 7 and Material and Methods.

Regarding the second comment, the tension sensor (TS) did not aim to replace Filamin-A in U2OS cells. TS used sequences from Filamin-A to create a shorter and flexible actin crosslinking protein to visualize forces within actin networks. The sensor lacks many domains present in FlnA and is not expected to fulfill all of the same functions. However, we recognize that expression of the TS introduces extra connectivity within actin networks, which may perturb those networks. These perturbations can manifest as changes in actin architectures, consequently impacting cellular forces. Therefore, we have addressed this by quantifying the actin alignment order (Material and Methods), and the magnitude of traction forces as a function of sensor expression level (Supplementary Figure 9). We do not see significant changes in these parameters; therefore, we suggest the sensor's perturbation is negligible in our analysis. This is mentioned in the main text now:

“The sensors are expressed in U2OS cells (Materials and Methods). We recognize that expression of the TS may perturb the actin network and its connectivity, which can manifest as changes in actin architectures and bundling, consequently impacting cellular forces. To assess the potential changes in the F-actin architecture, we measure the alignment of the F-actin ($\langle\varphi\rangle$, Material and methods), and to assess cellular forces, we measure the associated traction forces (Material and Methods) in TS cells (Supplementary Figure 9). We note that neither significantly change as a function of sensor expression, suggesting the tension sensor does not significantly perturb the cytoskeleton. However, CTS expressing cells show a higher

amount of diffuse sensor, compared to TS expressing cells, explainable by the lack of dimerization domain in the CTS structure which was also observed in non-patterned cells (Supplementary Figure 2).”

And in the discussion:

“First, we note that the tension sensor is not functionally equivalent to FLMN-A crosslinker. Rather, we used the actin binding domain (ABD) of FLMN-A to develop a sensor that binds to both disordered actin and stress fibers and reports forces between actin filaments within these structures. As the TS binds to two actin filaments with a preference for bundled and branched structures, the measured forces may indicate the magnitude of forces acting either perpendicular or parallel to the direction of the actin filaments, representing the effective force. The axial forces include resistance force to filament sliding within stress fibers, or axial forces along a single actin filament.”

- Although the intracellular stress inference is a very interesting approach, the results suggest that it is not relevant as it does not take the mechanical configuration of the cell into account. I suggest pointing this out even more pronounced than already done.

We have made this point clearer the in the discussion.

“Thus, the inferred intracellular stresses based on HIE assumptions vary for anisotropic cellular shapes fails as it does not take into account the internal mechanical configurations of the cell. We hypothesized that the mechanical anisotropy arises from the differential organization of SFs within distinct cell morphologies, and the extent of mechanical coupling between SFs and cortex.”

- I got confused with the way the order parameter was calculated. Unfortunately, although the explanation seems to have been due to be included in the Materials and Methods (line 298), it was not explained there. Does the local parameter refer to 5µm around the center of the cell? Or was the center of the 5µm radius region moved over the cells? In this case, what happens when the center of this point is at the cell edge? The same accounts for the global order parameter.

Thanks for raising this issue. We have now added to the Materials and Methods section:

“To quantify the alignment of the internal actin cytoskeleton, the alignment order was calculated in MATLAB®. After applying a Gaussian filter on raw images, the image is binned to small windows of 16z pixels, with 50% adjacent overlaps. A 2D Fast Fourier Transform (FFT) was then applied to all the windows, and the axis of the least second moments was calculated to determine the angle of the orientation in each window. Next, the local and global degrees of alignment were calculated according to:

$$\langle \phi \rangle = 2 \langle \cos^2 \delta\theta - \frac{1}{2} \rangle, \quad \delta\theta = \theta_i - \theta_j,$$

at a window centered to the point x_0 and y_0 , where $\delta\theta$ is the angle difference of adjacent vectors within a window size of r , i.e., $(i, j) = ([x_0 - r/2, x_0 + r/2], [y_0 - r/2, y_0 + r/2])$. The windows at the boundary of the cell were similarly analyzed, instead by ignoring the NaN values outside the cell area where there is

no alignment field. Thus, $\langle \varphi \rangle$ was calculated by taking the mean of $\langle \varphi \rangle$ over the entire area of the cell. We used $r = 5 \mu\text{m}$ and $r = 10 \mu\text{m}$ for the mean local and global order within the cell cytoskeleton.”

- The authors simply assume that all regions without stress fibers are dominated by cortical actin. What about intracellular actin structures? Or asked differently, is it valid to only separate the actin signal into stress fibers and cortex without including intracellular networks?

We understand that the reviewer is asking about the actin in the center of the cell distant from the plasma membrane. We use spinning disk confocal microscopy for imaging, which chiefly captures photons coming from a thin $\sim 500 \text{ nm}$ slice while blocking other parts of the sample. We do not expect to image the center of the cell (of heights $\sim 5 \mu\text{m}$) containing intracellular networks in our images. However, we may capture some of the intracellular networks adjacent to the basal surface, which is captured systematically in our segmented stress fibers and cortical actin. We have now added a line highlighting this.

“As we use confocal microscopy which captures a thin $\sim 500\text{nm}$ z-slice, we expect to image predominantly cortical actin, stress fibers, and intracellular networks near to the basal surface, but not the center of the cell. Thus, henceforth, we refer exclusively to stress fibers (SFs) and cortical actin. ”

- Do the normalized values of sensor enrichment in Figure 1E,F make sense? It will light up even at little TS in regions with little actin, so that unbound TS background would be mistaken as highly enriched regions. Why is this shown? Is not a correlation better? What does the I on the y-axis of Fig 1.G and H stand for? The short reference to this in the caption is a bit confusing.

The correlation only measures if the sensor is at the right location, but it does not measure how much of it is there. This analysis shows that the sensor binds to the majority of the actin structure, with different densities. Sensor enrichment can be important as it can introduce an excessive stiffness to the actin network. We have redone the analysis by omitting pixel values less than 30% of the mean and added the results as Supplementary Figure 4. The omitted regions are shown with dark blue (same as the background), and the reanalysis shows similar results. Therefore, the regions with a low amount of sensor do not significantly change the results. We have changed the I label to ρ (sensor enrichment) to avoid confusion and revised the captions. We have also referred to Supplementary Figure 4 in the text.

- It seems that the cell shown in video 2 contracts at timepoint 1:44, hence after the addition of calyculin (which makes sense). Should this not lead to an artificial increase in average traction as the area is decreasing? Effectively, it could lead to a situation where the TFM does not really increase in absolute values. To which extent does this influence the analysis?

As we have mentioned in the caption for Video 2, Calyculin-A was added at around 00:30, when the TFM heatmap starts to show increasing traction magnitudes while the cell keeps its initial area as visible from the ring, with a decrease in the average FRET index. The final contraction only happened at the end of the

movie when extremely high traction forces result in cell detachment. Therefore, we think this is reflecting an actual increase in the traction forces. Furthermore, previous works (doi.org/10.1016/j.bpj.2011.05.023) have shown up to 5-fold increase in traction forces in the same cell type (U2OS) upon treatment with 5 nM Calyculin-A. We have now updated video 2 and show how FRET efficiency changes after the addition of calyculin. We think the updated video 2 better represents the results.

- In Figure SI5 we see the quantification of Delta RFP, but it remains unclear how this was done or what it was used for. I expect that the total intensity of RFP is constant, so why should it change?
- The authors mention that the sensors partially unbind in parallel stretched cells referring to SI Fig 5. But how is this unbinding measured?

Thank you for pointing this out, we have now clarified this point in the updated Supplementary Figure 12, which includes the changes in the integrated intensity (total amount of sensor) and the average intensity (density of the sensor) of the sensor within the cells before and after stretching. We have updated the manuscript accordingly. Now it reads:

“Changes in the FRET E may also be due to the unbinding/binding of the sensor or appearance of cytosolic sensors in the focal plane. The quantification of the binding/unbinding of the sensor was performed by measuring total amount of the sensor ΔRFP and the density of the sensor $\Delta RFP/A$, where A is the cell area. With $\Delta RFP \approx 0.05$, we verified that in orthogonal cells, despite a basal level of sensor unbinding, the sensor remains bound during stretching (Supplementary Figure 12). Decreasing $\Delta RFP/A$ (≈ 0.10 , ≈ 0.16 , ≈ 0.21 , ≈ 0.31 for at 5%, 10%, 15% and 20% strain), further verified that the density of the sensor decreases with strain, as the area increases with strain. However, in parallel cells there is an increase in the density ($\Delta RFP/A \approx 0.2$) and the total amount of the sensor ($\Delta RFP \approx 0.2$), suggesting a deformation-induced revealing of binding sites on actin or integrins^{66,67} (Supplementary Figure 12). As this effect is not observed in the orthogonal direction, we suggest the elongation of actin-rich stress fibers which also colocalize with integrin sites reveals the majority of binding sites for the new sensors to attach.”

- The stop of FRET signal at 20 μ m Blebbistatin for any applied external strain is worrying me. This suggests that the FRET sensor depends on Myosin. The way the authors introduce the sensor suggests that they should bind to neighboring filaments, and these should still be stretched when the cell is stretched. Potentially, the sensor depends directly on myosin, and then we would measure not the force between actin but between actin and myosin. Here the functional test of the sensor as suggested above as control experiments would be important.

We appreciate the reviewer’s insightful comment. The sensor is an actin crosslinker, therefore it should measure the force between actin filaments. However, there are many other types of actin crosslinkers together with myosin that determine the force state of the cytoskeleton. Myosin, other than the force-generating role, has been identified as a crosslinker, which binds to actin filaments leading to the formation of interconnected structures. Blebbistatin hinders the force generation and crosslinking function (<https://doi.org/10.1007/s00424-007-0419-8>) of myosin which eventually softens the overall mechanics.

Unlike Blebbistatin, Y27632 has not previously been shown to influence cell stiffness (<https://doi.org/10.1007/s00424-007-0419-8>). Therefore, we have added additional Y27632 control, which is a ROCK inhibitor that inactivates Rho, an upstream and indirect inactivator of myosin, functionally similar to ML-7. We observed Y27632 treated cells exhibited consistent results to ML7 (Fig 4). We did not observe a force response for parallel cells however we did observe a switch in force for small strains. The

switch in both treatments is explainable by transverse resistive forces or inhibition of stretch-activated contraction through MLCK. Therefore, we decided to move myosin stretching results in the supplement, as we believe blebbistatin not only inhibits the force generation function, but also the actin-myosin binding interaction, which enables perturbation of the cytoskeleton under mechanical stretches (Supplementary Figure 13)

Additionally, intermolecular FRET control shows that it has a minimal effect on our calculations. (Supplementary Figure 7)

We have mentioned this in the main text

“We alternatively inhibited myosin activity with 10 and 20 μM Blebbistatin, which showed a dose-dependent and relaxation response at high strains (Supplementary Figure 13). However, cells detached from the substrate at high 20% strain, when treated with 20 μM Blebbistatin, suggesting extreme disruptions to the cytoskeleton or cell-ECM couplings. As Blebbistatin is known to directly perturb the F-actin cytoskeleton⁶⁰ and myosin’s actin-binding affinity⁶⁴, we expected disruptions to the cytoskeleton at high strains. However, in the absence of external strains, we measured the Blebbistatin-induced changes in internal and traction forces as shown in the previous section, despite possible cytoskeletal perturbations.”

And in the discussion:

“Parallel cells under ML7 or Y27632 treatments did not show a significant change in the internal forces, indicating the force-generation and resistance role of myosin in the observed anisotropy. Orthogonal cells under ML7 or Y27632 treatments, however, exhibited tensile forces compared to non-treated cells, suggesting stretch-activated binding of myosin II⁶³, or regulation of the cytoskeleton through MLCK⁶¹ (Fig 6B and 6C) to resist the deformation. As also previously shown in the context of cell stiffness, unlike Y27632, Blebbistatin induces softening in the cell cytoskeleton⁶⁰. Blebbistatin treated cells showed a relaxation response at high strains explainable by fluidization⁷⁰ due to direct inhibition of myosin ATPase and thus its binding to actin (Fig. 6B). At 20 μM Blebbistatin, the sensor exhibited little resistance against applied forces.”

- End of the results section, the authors use the found correlation between the SF length and the FRET signal. It is not clear how the total length is measured. Moreover, the authors take a strong statement from this correlation by stating that the amount of SF regulates the changes in stress response to strain. A simple correlation is not enough to infer causality. Please provide further data or remove this overstatement.

- Figure 5H, seems to be inconsistent with Figure 4G where the FRET values go to numbers exceeding 0.2.

Upon reanalysis of the data with FRET efficiency, we no longer see a correlation between the length of the SF and the change in FRET (shown below). The previous results might have reflected the intensity of the sensor as it is highly dense in the SFs, and that would also be reflected in the change of FRET, as we showed the FRET index is highly sensitive to the sensor expression (Supplementary Figure 9). However, FRET efficiency is not sensitive to the amount of sensors, and thus we updated our results. Therefore, we have removed those data from the paper. We thank the reviewer for this comment.

- Why do the authors not refer to figure 6 somewhere in the text?

Thanks for noticing the referencing issue. We have updated Figure 6 to include the inhibition data and the reference to Figure 6 is now added to the discussion.

- The authors use a UV illumination of the PAA to activate the patterns. Unfortunately, the authors do not provide information about the light dose and wavelength. In my experience, deep UV illumination of PAA to pattern the surface leads to changes in elasticity because the deep UV breaks chemical links. Did the authors quantify the stiffness of the gel at the patterns, e.g. by AFM?

We agree that the PAA gel stiffness can be altered by UV exposure for micropattern generation. We use a 2.8 kPa gel and 120 seconds of exposure to deep UV. Previous studies (<https://doi.org/10.1155/2017/5147482>) have shown that gel stiffness only slightly increases with ~100seconds of UV exposure (Figure 3b in the mentioned paper). Therefore, we understand UV's effect on the stiffness for our experiments is not influencing our results. We acknowledge this in the methods section.

- Overall, the authors report many average values without showing any error values. It would be important to also give an estimate of the validity of the mean value of TFM and FRET, for example by also showing the standard deviation.

Data shown in Fig 2C, D, F, are examples of single cells under drug treatment. Therefore, they do not contain standard deviation. We show the statistics of TFM-FRET relationship in Fig 2I and J where the mean and SD were calculated on multiple samples.

Minor:

- Line 103. Are the authors referring to Fig. 2D? There is no dashed line in 2E

Fixed.

- Line 114: Typo, “compared fluorescence the intensity” -> “compared the fluorescence intensity”

Fixed.

- SI Fig 4B, please define how the correlations are calculated. The definition of the order parameter remains unclear.

Added reference to the Materials and Methods

- Line 144: Typo “...to their direction of cell polarization” -> “...to the direction of cell polarization”

Fixed.

- Typo: Laica -> Leica

Fixed.

Reviewer #3 (Remarks to the Author):

SUMMARY

The authors created a novel FRET-based tension sensor based on a shortened version of filamin. The authors then attempt to use the sensor to learn about the role of anisotropy in intracellular mechanics. The comparison of traction forces and intracellular stresses in an important, but understudied, aspect of cell mechanics. However, the manuscript uses imprecise terminology when comparing molecular and macroscopic mechanical variables, lacks many controls for FRET-based tension sensors, and some of the calculations involving the interpretation of FRET values have errors (e.g. incorrect formulae). These major errors must be remedied before the manuscript can be fully evaluated.

The reviewer's comments have elevated its overall quality to a remarkable extent. Their constructive criticism and thoughtful recommendations have prompted the authors to delve deeper resulting in a more rigorous manuscript. The authors are grateful for the reviewer's rigorous evaluation.

We have done our best to address their comments, which includes designing new constructs for the suggested intermolecular and efficiency controls, and based on that, we have reanalyzed the entire data in the paper to reflect FRET efficiency. **This has increased our confidence in the results and would like to note that our salient conclusions are unchanged. Thus, we have replaced the FRET index results, or moved them to the supplement, depending on the context.**

MAJOR CONCERN

1. The authors switch between using mechanical terms that are best defined at large length scales in two or three dimensions where continuum mechanics approaches (e.g., stress, strain...) are appropriate and a molecular description that is appropriate for describing the FRET-based tension sensor (tension, compression...). This needs to be resolved to prevent many potential issues in interpretation of the results. In this review, I will place the word "molecular" before mechanical variables when I am referencing the sensor. I suggest the authors first be more precise in the use of mechanical terminology across length scales and adopted a similar convention of their choice.

We thank the reviewer for this important comment. We have revised the terminology. We believe this will make our contribution even more significant in showing that intracellular forces at the molecular level are related to the externally applied tractions in the TFM measurements. We chose to follow the typical way of presenting the TFM results by showing the stresses, however, traction vectors can be represented in terms of force as well.

2. It has been reported that the molecular tension sensors can report molecular compression (i.e. a negative

tension in a one dimensional sense) (see doi: 10.1038/nature13535 and 10.1007/s12195-015-0404-9). Thus, it is critical to establish a value of FRET corresponding to the unloaded state. FRET values less than this value are consistent with the sensors experiencing molecular tension (on average, within an imaging voxel). Higher values are consistent with molecular compression of the sensor and/or the onset of intermolecular FRET (more on this below). Thus, it is very problematic that unnormalized FRET index values shown in Fig 2J are so low for the presumably unloaded CTS. Specifically, the presumably unloaded CTS values have FRET indices of approximately 0.15. The FRET values for TS at low traction forces are as high as 0.6. This would suggest that either CTS is not functioning as expected or significant molecular compression in the sensor and/or the presence of substantial intermolecular FRET in the TS. Currently, these discrepancies render all FRET data in the manuscript uninterpretable, and this issue must be addressed before publication.

We thank the reviewer for bringing this constructive comment. For Fig 2, we have done the following:

- We noticed that our original data in Fig 2J comprised cells that varied in the expression level of the sensor as we had a population of transiently (older data) and stably transfected cells. Now in Supplementary Figure 9, we show that there is a larger variance in the expression level of transiently transfected cells.
- We also show that the sensor expression influences the FRET index – which may induce an artifact in the absolute values of the FRET index. However, FRET efficiency is not a function of sensor expression (Supplementary Figure 9). This issue was resolved by carefully choosing the cells that have an average intensity of around 500 and are stably transfected where there is not much variance in the basal level of the FRET index (panel D)
- We updated the old Fig 2J (now in Supplementary Figure 9D), and the issue is resolved, as the FRET index values for TS in low force correspond to that of CTS.
- Please also note that we have swapped the x-y axes to avoid having a vertical line for the CTS data.
- Based on your later comment, we have conducted the necessary experiments to calculate the FRET efficiency and updated the entire Fig 2 with FRET efficiency results. (Details in the response to the related comment.). We show that CTS shows a $\langle \text{FRET E} \rangle \approx 22 \pm 2\%$ while TS ranges from $\approx 33\%$ at low forces, down to $\approx 15\%$ at high forces.
- Given that TS+blebb FRET efficiencies values can be larger than that of CTS in the 25-30% range, we have conducted a significance test to determine whether the TS shows an average molecular compression. T-test of $\langle \text{FRET E} \rangle$ for low stresses ($\langle |T| \rangle$ less than 50 (Pa)) between TS+blebb and CTS cells showed a significant difference $P < 0.001$, indicating the existence of molecular compression on average over the cell area. This was also confirmed by comparing the FRET E of lysed TS and CTS cells to the TS+Blebb. cells. We have added this in the main text now.

3. One possibility for the discrepancy in FRET values for the TS and CTS is that the TS is a dimer and undergoing intermolecular FRET. This would drastically affect the interpretation of the data. As has been specified in multiple reviews on the creation and use of FRET-based tension sensors, the authors should perform intermolecular FRET controls involving dark mutants of the various FPs (e.g., 10.1007/s12195-014-0368-1). Notably, this approach was pioneered in the Schwartz Lab (10.1016/j.cub.2013.04.049). In the opinion of this reviewer, this control must be performed for the manuscript to be suitable for publication.

In light of the reviewer's insightful comment, we have conducted intermolecular FRET controls and added them to the manuscript. We adopted the method in (10.1007/s12195-014-0368-1). Briefly, point mutagenesis was done using designed primers in Supplementary Table 1, whose method is added to the Material and Methods. Therefore, we made two new TS constructs in which either RFP or GFP is dark mutated.

Co-expressing these constructs in dark U2OS cells using transient transfection, we verified that the intermolecular FRET is negligible. Now in Supplementary Figure 7, we show that the intermolecular control shows a FRET efficiency as low as, and insignificantly different from, the low FRET efficiency TS-TRAF construct. This comparison shows that the **intermolecular FRET is negligible** as its FRET efficiency is comparable to the long linker (TRAF) construct. We have added this to the main text.

4. Furthermore, the authors have chosen to represent the majority of their FRET data as “change in FRET”. This prevents discerning of the absolute values of the FRET index. This leads to problems in the conceptualization of the results, as in the context of FRET-based tension sensors “molecular compression” has been used to describe sensors are pushed closer together than their unloaded resting state. The context of “compression” in this manuscript is not immediately clear to this reviewer. I think, but I am no 100% sure, that the authors are using “compression” to mean “increase in FRET” from a loaded baseline. Thus, “compression” could refer to reduction of tension (e.g. from high to low molecular tension, instead of the onset of molecular compression). Furthermore, the relationship between FRET index and molecular tension is not linear and is unknown for molecular compression (see <https://doi.org/10.7554/eLife.33927>). Thus, it is unclear what the change in FRET Index metric actually represents (see below for more regarding the calculations in the supplement). The terminology issue, as well as an inability to discern the absolute value of the FRET indices, should be fixed to enable interpretation of the data in the manuscript.

We thank the reviewer for bringing this unclear point to our attention. The FRET efficiency analysis has now enabled us to compare between cells. We verify that in the stretching experiments, the FRET efficiencies do not go to the compressive (Supplementary Figure 12) regime. We interpret the positive change in the FRET efficiency to be a relaxation or compression from a loaded baseline. Now we have clearly stated this in the text.

5. The idea of comparing the traction forces, internal “stresses” determined from inference methods, and the FRET signal is interesting. But, it is unclear this analysis can be interpreted for multiple reasons. To begin, I will discuss the mechanical issues. It is completely unclear if the sensors report all internal stresses in the cell. First, there are a plethora of endogenous actin cross-linking proteins. These could all be bearing portions of the intracellular stresses not detected by molecular tension sensor. Secondly, the sensor probably estimates forces between F-actin filaments, and not forces transmitted along filaments, which might be expected to dominate in stress fibers. Thirdly, as shown in Figure 1, the authors demonstrate that the sensor does not equally label all actin structures. Thus, components contributing to external stress field could be missed by the sensor. Next, I will discuss it from the sensor side. There are many issues due to the dynamic binding of the sensor. As shown in the supplemental material, the binding time of the sensors is quite short. Thus, sensors could be bound to no filaments (i.e. soluble) or bound to a single filament. The relationship of sensors in these states to stress measurements is unclear. Also, there are issues with the sensitivity of the

tension sensing modules. The authors used FRET Index for this comparison. FRET Index is not simply related to FRET Efficiency (10.1529/biophysj.106.088773). Furthermore, FRET Efficiency is non-linearly related to the underlying molecular tension (10.1038/nature09198, <https://doi.org/10.7554/eLife.33927>). It is unclear how not accounting for these complex relationships will affect the comparison between FRET signals, intracellular stresses, and traction forces. Additionally, the proposed analysis requires that the sensors accurately report the actual molecular tension that it reports. This sensitivity of the sensor is like in 1-6 pN range, and molecular forces outside that range cannot be measured. Furthermore, no sensor has been calibrated for molecular compression. Thus, if there are sensors in molecular compression, how their FRET signal contributes to estimates of mechanical stress is completely unclear. Additionally, any contribution from intermolecular FRET would bias these comparisons as well. For all these reasons, this reviewer suggests that figure 3 be re-done with a more correct analysis (see below, potentially involving FRET efficiency if concerns regarding intermolecular FRET and molecular compression can be addressed) and then added as supplemental figure. I expect that the inferred internal stresses will not match the tension sensor signal, but, due to the issues listed here, it is unclear how to interpret that result. However, it clearly cannot be interpreted as due to a lack of consideration of anisotropy alone. Furthermore, these inference methods also ignore all dissipation of mechanical energy of the system as well as local variation in the mechanical properties of the system.

We would like to thank the reviewer for this insightful comment. To address the comment effectively, we have broken it down into bullet points and provided responses to each point individually.

- “First, there are a plethora of endogenous actin cross-linking proteins. These could all be bearing portions of the intracellular stresses not detected by molecular tension sensor.”

First, we would like to clarify that our intention was not to claim that we measure all internal stresses within the cell. We agree that there are numerous endogenous actin cross-linking proteins that may bear portions of the intracellular stresses, which would not be detected by our molecular tension sensor. We believe that our molecular tension sensor reports the relative forces between actin filaments. These forces may be in the axial direction resisting the sliding mechanism, tension in one single filament, or transverse to the bundles. As this sensor binds to actin all over the cytoskeleton in the presence of other crosslinkers, we believe our measurements still reflect the force distributions and variations. This can be thought of as a systematic error in the measurement method which does not significantly influence our findings.

- “Secondly, the sensor probably estimates forces between F-actin filaments, and not forces transmitted along filaments, which might be expected to dominate in stress fibers.”

Regarding the second point about the sensor estimating forces between F-actin filaments rather than forces transmitted along filaments, we acknowledge the importance of considering the dominant forces in stress fibers. Depending on the orientation of the sensor, it can represent a combination of axial forces (by force balance along actin filaments) and transverse forces. The axial forces can either be resistive to filament sliding mechanism within a stress fiber, or axial forces in one single actin filament. Based on our analysis, these forces are non-zero and change within stress fibers and cortex, without any assumption. Regardless, our analysis takes into account both components, as the total force. We believe it provides valuable information about the forces present in the cell.

- “Thirdly, as shown in Figure 1, the authors demonstrate that the sensor does not equally label all actin structures. Thus, components contributing to external stress field could be missed by the sensor.”

We agree with your observation that our sensor does not equally label all actin structures, as demonstrated in Figure 1. We acknowledge that there might be components contributing to the external stress field that could be missed by the sensor. However, the strong and significant Pearson's correlation we observed between the sensor and actin localization indicates that the sensor is predominantly present where actin is. This suggests that the sensor is capable of measuring forces at locations where it binds to actin. Of course, in regions with lower actin density, the probability of force transmission may be attenuated. We further show that sensor expression level does not influence the FRET E (Supplementary Figure 9), This assurance allows us to conclude that changes in concentration would not impact our measurements.

We appreciate your insightful comments, and we have ensured to emphasize these limitations and considerations in our revised manuscript. We believe that despite these limitations, our study contributes to our understanding of cellular forces and provides a valuable framework for further investigations.

- “Next, I will discuss is from the sensor side. There are many issues due to the dynamic binding of the sensor. As shown in the supplemental material, the binding time of the sensors is quite short. Thus, sensors could be bound to no filaments (i.e. soluble) or bound to a single filament. The relationship of sensors in these states to stress measurements is unclear. Also, there are issues with the sensitivity of the tension sensing modules. The authors used FRET Index for this comparison. FRET Index is not simply related to FRET Efficiency (10.1529/biophysj.106.088773). Furthermore, FRET Efficiency is non-linearly related to the underlying molecular tension (10.1038/nature09198, <https://doi.org/10.7554/eLife.33927>). It is unclear how not accounting for these complex relationships will affect the comparison between FRET signals, intracellular stresses, and traction forces. Additionally, the proposed analysis requires that the sensors accurately report the actual molecular tension that it reports. This sensitivity of the sensor is like in 1-6 pN range, and molecular forces outside that range cannot be measured. Furthermore, no sensor has been calibrated for molecular compression. Thus, if there are sensors in molecular compression, how their FRET signal contributes to estimates of mechanical stress is completely unclear. Additionally, any contribution from intermolecular FRET would bias these comparisons as well. For all these reasons, this reviewer suggests that figure 3 be re-done with a more correct analysis (see below, potentially involving FRET efficiency if concerns regarding intermolecular FRET and molecular compression can be addressed) and then added as supplemental figure. I expect that the inferred internal stresses will not match the tension sensor signal, but, due to the issues listed here, it is unclear how to interpret that result. However, it clearly cannot be interpreted as due to a lack of consideration of anisotropy alone. Furthermore, these inference methods also ignore all dissipation of mechanical energy of the system as well as local variation in the mechanical properties of the system.”

In light of the reviewer's comment, we conducted new experiments to enable the calculation of the FRET efficiency of the sensor. We have made new constructs of TS in which the F40 linker is replaced with short (GGSGGS)₂ or long (TRAF) linkers, to get the highest and lowest FRET efficiencies, respectively. We used the method described in <https://doi.org/10.1529/biophysj.106.088773> to calculate the FRET efficiencies (FRET E). The <FRET E> for the short linker is about ≈ 33%, comparable to other reported

values (<https://doi.org/10.1002/cyto.a.23688>). Our measurements of FRET efficiency range between $\approx 12\%$ to 33% and this is the functional range of force measurement by our sensor.

Figure 3 was redone and added as Supplementary Figure 10. In the updated figure, we observe consistent results with the older results. We chose FRET efficiency over the FRET index as the main measure of the FRET signal.

6. There are several errors in the Supplemental Note 2. The molecular tension sensors do not act like linear springs in the cell domain, and the referenced citation does not show that. The extensible domains of the tension sensors act like unstructured polypeptides that can be well described using models of worm-like chains from polymer physics. See <https://doi.org/10.7554/eLife.33927> for details. Secondly, the equation relating FRET index and FRET Efficiency is incorrect. These variables are not directly proportional, and the cited reference does not state that they are. The relationship between FRET efficiency, the various imaging channels, and the correction factor G can be found in Chen, Biophysical Journal, 2006 (<https://doi.org/10.1529/biophysj.106.088773>). It is not clear to this reviewer what if any data these errors effect, but they should be corrected and effected analyses re-done. The errors do seem to affect how the authors are interpreting their data. For instance, I believe these calculations are the basis for interpreting the change in FRET in terms of mechanical variables. Thus, the interpretation of the data and discussion should be re-considered once the errors are corrected.

Thanks for highlighting the errors in Supplementary Note 2. We have completely reanalyzed Figure 4 and Figure 5 to reflect the FRET efficiency measurements. The purpose of Supplementary Note 2 was to show a relationship between the FRET index and force. However, now that we have updated our results in terms of FRET efficiency, we have removed the majority of Supplementary Note 2, and have relied on the absolute and relative changes in the FRET efficiency to conclude about intracellular forces under mechanical strains.

7. If the authors have used “three-cube” FRET imaging as stated in the methods, then they should be able to readily calculate FRET efficiency. This is described in <https://doi.org/10.1529/biophysj.106.088773>, and has been used in the context of FRET-based tension sensors (<https://doi.org/10.7554/eLife.33927>, <https://doi.org/10.1002/cyto.a.23688>). These FRET efficiencies can be converted to the molecular tension experienced by the sensor if the tension sensing module has been suitably calibrated.

We would like to thank the reviewer for providing useful references. We have calculated FRET efficiencies from our data and updated all the figures accordingly.

The sensing module has been calibrated previously (<https://doi.org/10.1038/nature09198>), however between a different pair of fluorophores. Although different pairs of FPs exhibit different Forster radii, the difference is not large, so we could use the calibrated tension and efficiency conversion. However, as the forces within the actin network can be in any direction, a direct FRET to force conversion may not reflect the real force state. In contrast, talin/vinculin FRET tension sensors report forces that are predominantly normal to the cell-substrate interface. Therefore, we decided to use FRET efficiency as a proxy for force without force conversion. We have mentioned this in the main text now.

8. The authors first state the slope (m) shown in Figure 5, can interpreted as indicating the degree of load bearing by the various structures. I agree with this interpretation. However, they offer an alternative interpretation that says that the data can be used to infer the stiffness of the structures. This later interpretation seems to assume that the two structures are subject to same mechanical input. However, this is not necessarily true as the stretching forces are transmitted to the various cytoskeletal elements through adhesion receptors and coupling between the two structures and the adhesions may not be uniform. It is opinion of this reviewer that either experiments directly probing the stiffness of the various actin structures be added to the manuscript or this alternative interpretation be removed from the paper. Specifically, without additional data, Figs. 5 J and K need to be re-stated in just terms of differential load-bearing.

We agree with the reviewer regarding non-uniform strain applied to the intracellular structures. Therefore, we removed our interpretation of the slopes to the ratios of stiffness. Instead, we simply call it “SF-Cortex load bearing”. In the updated Fig 5, FRET efficiency shows that in the orthogonal direction, the stress fibers and cortex bear undergo relaxation and tension, indicative of differential load bearings. This was also reflected in the positive versus negative slopes of Fig 5D. This behavior was not observed in ML7 or Y27632 controls, as both showed a positive and similar slope of ~0.7. Therefore, we show that myosin changes the direction of load bearing (tensile/relaxation), however upon myosin inactivation, stress fibers and cortex always bear loads to different extents, although in the same direction. We have updated the text to reflect this.

9. I believe the main point of Fig.6 is that there is an unexpected increase in tension in the orthogonal direction for large strains. This effect does not appear to occur in blebbistatin-treated cells (Fig.4I). Furthermore, the response is reversed if ML-7 is used to inhibit MLCK, and the expected increase in FRET is observed. I believe this suggests a counter hypothesis that at high strains the cells activate myosin activity (likely through MLCK) to globally increase molecular tension. This alternatively hypothesis should be excluded or at least discussed. One suggested (but not required) experiment would be to address the role of stretch activated calcium channels in this system, which could be activated at large deformations. MLCK is associated with Calcium-based regulation of contractility, so this connection could explain the primary observation. Regardless of the additional experiments, this reviewer suggests that the inhibitor data be included in the summary Figure 6 and the statement of the primary results of this manuscript.

We thank the reviewer for suggesting an alternative hypothesis and insightful comment. ML-7 results and now supported by the added control of Y27632. We have included this hypothesis in the discussion besides the stretch-activated binding of myosin II to actin as another hypothesis. Based on the new results, we have updated Fig 6 as well.

MINOR CONCERNS

1. In Fig. 2C it is not immediately clear what is shown, as “Control” could be in reference to the use of CTS, the lack of an inhibitor treatment, or both.

We have rearranged and updated Fig 2. The no drug control is in Supplementary Figure 8 with appropriate labeling.

2. The text states that “We verified that the sensor remains bound in orthogonal cells, however, it partially unbinds in parallel cells (Supplementary Figure 5).” The opposite result seems to be shown in that Figure. Also, it seems that the error bars are much larger in C than in B. This could be important as the size of the effect in C is often the same as in B. The authors should elaborate on the origin of the different variances as well.

Thank you for pointing this out, we have now clarified this point in the updated Supplementary Figure 12, which includes the changes in the integrated intensity (total amount of sensor) and the average intensity (density of the sensor) of the sensor within the cells before and after stretching. We have updated the manuscript accordingly. Now it reads:

“Changes in the FRET E may also be due to the unbinding/binding of the sensor or appearance of cytosolic sensors in the focal plane. The quantification of the binding/unbinding of the sensor was performed by measuring total amount of the sensor ΔRFP and the density of the sensor $\Delta RFP/A$, where A is the cell area. With $\Delta RFP \approx -0.05$, we verified that in orthogonal cells, despite a basal level of sensor unbinding, the sensor remains bound during stretching (Supplementary Figure 12). Decreasing $\Delta RFP/A$ (≈ -0.10 , ≈ -0.16 , ≈ -0.21 , ≈ -0.31 for at 5%, 10%, 15% and 20% strain), further verified that the density of the sensor decreases with strain, as the area increases with strain. However, in parallel cells there is an increase in the density ($\Delta RFP/A \approx 0.2$) and the total amount of the sensor ($\Delta RFP \approx 0.2$), suggesting a deformation-induced revealing of binding sites on actin or integrins^{66,67} (Supplementary Figure 12). As this effect is not observed in the orthogonal direction, we suggest the elongation of actin-rich stress fibers which also colocalize with integrin sites reveals the majority of binding sites for the new sensors to attach.”

3. The authors should specify which type of blebbistatin that was used for live cell imaging. Classical blebbistatin can be toxic in live cell assays and can lead to substantial background signal in FRET experiments.

We used (-)-Blebbistatin (<https://www.sigmaaldrich.com/US/en/product/sigma/b0560>) from Sigma Aldrich, which has applications in live cell imaging. We added this to Material and Methods.

We realized Blebbistatin results could be reflecting the change in the cytoskeletal structure as Blebbistatin directly influences the actin-myosin interaction and binding, which can be amplified under mechanical strains. We, therefore, decided to move the stretching results with Blebbistatin to Supplementary Figure 13. Instead, we have taken your later recommendation to include ROCK inhibitor (Y27632) in the results in which we observed consistent results with ML-7 control. Both Y27632 and ML-7 indirectly inhibit myosin activity implying less perturbation to the actin network.

RECOMMENDATIONS

4. There are many experimental conditions but relatively few pictures of cells. In the context of Fig. 5, it would be nice to have more pictures of the cells to create a clear picture for the reader. For instance, do the cells treated with blebbistatin or ML-7 have altered stress fibers?

Thanks for the suggestion, we have now added images for cells related to each drug control. Blebbistatin cells seem to have altered, shorter stress fibers, whereas ML7 treated cells seem to be similarly containing stress fibers.

5. The paper would be strengthened with the inclusion of experiments addressing the role of RhoA-ROCK mediated myosin activation, as in addition to MLCK, this is the other major pathway of regulating myosin activity. Experiments with a ROCK inhibitor would be interesting.

Thanks for the great recommendation. As we mentioned in minor comment 3, we have added Y27632 control, a ROCK inhibitor to the stretching experiments in Fig 4 and Fig 5, which shows a response consistent with ML7 treated cells.

6. In the opinion of this reviewer, the most interesting data in the manuscript is that fact that there are multiple relationships between the FRET signal of the sensor and traction force, as shown in Fig. 2J. It would be good for the field to discuss this observation more.

In the updated Fig 2J, there is still significant differences between the FRET-TFM relationships. We have also discussed it in the results and discussion section.

REVIEWER COMMENTS

Reviewer #1 (Remarks to the Author):

The authors answered most of my comments faithfully except for the following point. I pointed out that references 36 and 37 are inappropriate to refer to RhoGEF. I will more clearly state the reason. The reference 36 by Hoffman, B. D., Grashoff, C. & Schwartz, M. A is an excellent review paper, but did not refer to Rho GEF very much. In the reference 37 by Doss et al., the term “GEF” appears only once as GEF-H1.

Reviewer #2 (Remarks to the Author):

I congratulate the authors on an excellent article. I am satisfied with the answers and support the revised manuscript for publication in Nature Communications. While reading, I found some small errors that the authors probably want to correct before publication. Line 235: there seems to be a typo. I do not understand what “...rem bound..” means. Line 440: Typo: “...and we the inverse masks...” Figure 1CD, I suggest increasing the brightness for the Label 'Paxillin'. On a standard laser printer the letters almost merge with the black background.

Reviewer #3 (Remarks to the Author):

SUMMARY: The authors should be commended for the substantial effort they have put into the response. However, there are still issues with the use of the FRET-based tension sensor. This includes how calibrations were performed, interpretation of controls, interpretation of experimental data, and general conflation of mechanical terms across scales. In the opinion of this reviewer, the manuscript is still not suitable for publication.

MAJOR CONCERNS

1) My first major concern in the previous review was not fully addressed. The authors are still conflating molecular tension and mesoscale/macroscopic force throughout the manuscript. FRET-based tension sensors do not measure force. Even at the molecular level, there is no vector information. At larger length scales, the sensors are only one element in complex network and how their output relates to continuum level variables, which reflect

all elements in the system, is not clear and cannot be readily defined. The authors are interested in studying the forces within the F-actin cytoskeleton. However, this is not the “mechanical information” a FRET-based tension sensor reports. The sensor reports the tension across the linker region between the two FPs in whatever molecular species it has been placed in. This is only readily related to the tension across the protein for simple geometries (like considering the sensor module and the rest of the protein as spring in series, as is likely the case for vinculin). The geometry of the sensor here (as a dimer) is much more complicated, and thus even more separation between the sensor output and continuum level variables is appropriate. To be clear, I have no problems with comparing the FRET signals to the traction forces qualitatively. This is a worthwhile endeavor, but the authors are over-interpreting the relevance of the mechanical information provided by the molecular tension sensor to continuum mechanics. For this manuscript to be suitable for publication, the authors must develop and use clear and separate terminology for when they are discussing molecular tension and macroscopic mechanical variables, as I stated in the last review. For instance, force is currently used to refer to a variety of drastically different things (e.g., traction vectors and the output of the tension sensor). These should have different terms and comparisons should be made very carefully and probably only qualitatively. To be clear, I am fine with the language the authors using to describe the traction force microscopy. They clearly have substantial expertise in this realm. But this should be clearly distinct from that used to describe the molecular tension sensors.

2) On a related note, the authors are attempting to use the use the molecular-scale and molecular-specific readout of the FRET-based tension sensor as a stand-in for an intracellular force field (i.e. a macroscopic mechanical variable). This comparison is made in Fig. 3 by comparing the inferred internal stresses and the FRET signal. I do not believe this is a valid/interpretable comparison. This is the equivalent to using the tension on the two arms of a Y-shaped sticks to infer the force on all stick in a giant pile. So, the calculation of the inferred stresses is useful, but the simplest interpretation is that the inferred stresses do not match the tension sensor data b/c one is inferred from the entire force generation of the cell and the other is based on a molecular measurement of one species of molecular crosslinker. While this comparison can be used to emphasize the complexity of the sensor signal, it should not be used to motivate the importance of anisotropy or any particular

mechanical variable. It is fine to talk about anisotropy or the presence of F-actin structures. But doing this comparison to motivate the importance of anisotropy when we know there are anisotropic stress fibers in most cells is unnecessary and promotes misuse and misinterpretation of the molecular tension sensor. For this manuscript to be publishable, the strong comparisons of macroscopic stresses and tension sensor output must be removed and replaced with less definitive qualitative conclusions.

Furthermore, I note that reviewer 1 made a similar comment. The fact that the inferred stresses don't match the tension sensor is also not surprising as the research field has known for decades that cells are not a linearly elastic homogenous solids. Thus, the result that inferred stresses do not match the molecular tension sensors is surprising for a variety of reasons and should not be strongly emphasized.

3) The color bars for the FRET in Fig.3 should be changed. Currently there is a correspondence between 0 Pa Stress and 30% FRET efficiency. However, the 0 pN state of the sensor is 22% FRET efficiency (more on the accuracy of this estimate below). So, if the authors still want to make this comparison, I guess 0 pN and 22% should be matched, at least? But it seems clear that 0 stress and molecular compression should not be matched. I note the vagary in how to set this color table to ensure a viable comparison is generally indicative of the difficulties in comparing molecular tension and macroscopic stress.

4) Estimation of G correction factor was done incorrectly due the specifics of the control constructs. Constructs that can be subjected to load cannot be used for this purpose. The premise of the calibration is that the FRET of the two constructs are different and constant. If the constructs can be loaded, the FRET is not constant, and the calibration cannot be performed. The calibration should be re-done with soluble versions of GFP-GGSGGS2-RFP and GFP-TRAF-RFP (meaning that there are no flanking domains), as has been done previously. Notably, there are differences between the calibration constructs based on whether they are in stress fibers or the cortex (Sup. Fig. 6 and 7), indicating the variability of the constructs used in calibration, and clearly demonstrating the need to re-do the calibration. For this paper to be publishable, the unloadable calibration constructs should be produced, used to determine the required calibration factors for determining FRET

Efficiency, and then the data reprocessed.

5) I disagree with the authors' interpretation that there is not intermolecular FRET in their system. The TRAF linker is designed as a low FRET control, not a no FRET control. 13% FRET efficiency likely indicates substantial intermolecular FRET. To resolve this issue the authors should co-express GFP and soluble RFP (preferably in a bi-cistronic vector to keep expression levels relatively similar) in their system. The only FRET that can occur in this experiment is bystander FRET. Thus, it will give you an estimate of how a system with very little intermolecular FRET performs. I would anticipate that FRET efficiencies will be less than 5%, a common noise floor in these types of measurements. For this manuscript to be publishable, more definitive evidence that there is a lack of intermolecular FRET in the system must be provided. Without it the contributions of molecular compression versus intermolecular FRET cannot be separated, and the experimental data cannot be interpreted.

6) The FRET seems substantially lower on the PDMS. A potential reason for this should be given in the text. Also, showing (or at least stating) that CTS sensors report the same FRET Eff on the PA gels used for tractions and the PDMS membranes used for stretch experiments would help alleviate concerns there are technical issues with the experiments. For instance, is it possible that G is different in the two set-ups? If so, G will need to be determined for both.

MINOR:

1) The authors sometimes refer to the sensor as a filamin sensor and sometimes a modified filamin sensor. This has generated substantial confusion amongst the reviewers. As the sensor is comprised of a relatively small portion of filamin, I might suggest a different name not involving filamin at all or more explicitly pointing out its small fraction. Also, how the sensor is referenced is not the same throughout the manuscript (sometimes it is just called TS). A single term should be used to reference the sensor developed in this work.

2) The author sometimes refer to the sensor as a force sensor. It should not be referred to as a force sensor, as no vectorial information is provided by the sensor. Thus, it should be referred to as a tension sensor.

3) Intermolecular FRET is not referred to as “crosstalk FRET”. In FRET crosstalk is most commonly used to refer to where the donor FP generates signal in acceptor channel and vice versa. This terminology should be removed.

4) In Supp Fig 6, the TRAF domain is drawn incorrectly. It is a structured domain, not a long flexible linker.

Reviewer #1 (Remarks to the Author):

The authors answered most of my comments faithfully except for the following point. I pointed out that references 36 and 37 are inappropriate to refer to RhoGEF. I will more clearly state the reason. The reference 36 by Hoffman, B. D., Grashoff, C. & Schwartz, M. A is an excellent review paper, but did not refer to Rho GEF very much. In the reference 37 by Doss et al., the term “GEF” appears only once as GEF-H1.

We appreciate the reviewer’s constructive and positive feedbacks and are glad to hear that the revised version was satisfying. As requested, we have now fixed and updated the references regarding RhoGEF:

- Hoffman, B. D., Grashoff, C. & Schwartz, M. A. Dynamic molecular processes mediate cellular mechanotransduction. *Nature* 475, 316–323 (2011).
- Mayer, M., Depken, M., Bois, J. S., Jülicher, F. & Grill, S. W. Anisotropies in cortical tension reveal the physical basis of polarizing cortical flows. *Nature* 467, 617–621 (2010).
- Motegi, F. & Sugimoto, A. Sequential functioning of the ECT-2 RhoGEF, RHO-1 and CDC-42 establishes cell polarity in *Caenorhabditis elegans* embryos. *Nat. Cell Biol.* 8, 978–985 (2006).

Reviewer #2 (Remarks to the Author):

I congratulate the authors on an excellent article. I am satisfied with the answers and support the revised manuscript for publication in *Nature Communications*. While reading, I found some small errors that the authors probably want to correct before publication.

We are delighted to know that our work has met the reviewer’s expectations. We appreciate your endorsement of the revised manuscript for publication. Below we have addressed the remaining errors.

Line 235: there seems to be a typo. I do not understand what “...rem bound..” means. **fixed.**

Line 440: Typo: “...and we the inverse masks...” **fixed.**

Figure 1CD, I suggest increasing the brightness for the Label ‘Paxillin’. On a standard laser printer the letters almost merge with the black background. **Fixed**

Reviewer #3 (Remarks to the Author):

SUMMARY: The authors should be commended for the substantial effort they have put into the response. However, there are still issues with the use of the FRET-based tension sensor. This includes how calibrations were performed, interpretation of controls, interpretation of experimental data, and general conflation of

mechanical terms across scales. In the opinion of this reviewer, the manuscript is still not suitable for publication.

We appreciate the recognition of our efforts in addressing the concerns raised in the response. We value the reviewer's constructive feedback and believe they have been helping us improve the quality of our manuscript. Here we have made our best effort to address the remaining concerns.

MAJOR CONCERNS

1) My first major concern in the previous review was not fully addressed. The authors are still conflating molecular tension and mesoscale/macroscopic force throughout the manuscript. FRET-based tension sensors do not measure force. Even at the molecular level, there is no vector information. At larger length scales, the sensors are only one element in complex network and how their output relates to continuum level variables, which reflect all elements in the system, is not clear and cannot be readily defined. The authors are interested in studying the forces within the F-actin cytoskeleton. However, this is not the "mechanical information" a FRET-based tension sensor reports. The sensor reports the tension across the linker region between the two FPs in whatever molecular species it has been placed in. This is only readily related to the tension across the protein for simple geometries (like considering the sensor module and the rest of the protein as spring in series, as is likely the case for vinculin). The geometry of the sensor here (as a dimer) is much more complicated, and thus even more separation between the sensor output and continuum level variables is appropriate. To be clear, I have no problems with comparing the FRET signals to the traction forces qualitatively. This is a worthwhile endeavor, but the authors are over-interpreting the relevance of the mechanical information provided by the molecular tension sensor to continuum mechanics. For this manuscript to be suitable for publication, the authors must develop and use clear and separate terminology for when they are discussing molecular tension and macroscopic mechanical variables, as I stated in the last review. For instance, force is currently used to refer to a variety of drastically different things (e.g., traction vectors and the output of the tension sensor). These should have different terms and comparisons should be made very carefully and probably only qualitatively. To be clear, I am fine with the language the authors using to describe the traction force microscopy. They clearly have substantial expertise in this realm. But this should be clearly distinct from that used to describe the molecular tension sensors.

We appreciate the reviewer for bringing up the concerns about the terminology used in our manuscript. In response to the reviewer's comments, we have made the following changes to the manuscript:

- **Clear Terminology:** To address the conflation of molecular tension and macroscopic force, we have introduced distinct terminology for microscopic and macroscopic quantities. Specifically, we now refer to the output of TFM as "**traction**" and the FRET signal as "**molecular tension**". This differentiation enables us to clearly distinguish between molecular and continuum-level variables in the manuscript.
- **Qualitative Comparisons:** It is noteworthy that we compared the FRET efficiency with the tractions to demonstrate a proof of principle that the tension sensor is capable of measuring mechanical tension *in-vivo*. It shows the balance of internal molecular tensions and external tractions, and **therefore indicates that the TS is functional and sensitive to forces, supporting**

the capability of the tension sensor to provide mechanical information. However, **we acknowledge that interpreting the direct quantitative comparisons between molecular tension measurements and macroscopic mechanical variables is not straightforward** due to reasons such complexity of the tension sensor's geometry, mechanical coupling to the ECM or viscous dissipations. As such, we have taken a cautious approach and limited our comparisons to qualitative assessments. Therefore, **we have removed words such as ‘benchmark’ to avoid misinterpretations.** For all these reasons, we have not utilized the results in Figure 2J as a calibration curve throughout the manuscript.

The text is now updated specifically to reflect our approach:

“It is important to highlight that the observed correlation between molecular tensions and tractions does not represent a quantitative calibration between internal and external forces as the probed molecular tension between actin filaments may not be directly translated into tractions reflecting the entire cytoskeletal structure. This may be due to the tension sensor’s dimerized geometry, mechanical coupling to the ECM, or viscous dissipations. Nevertheless, the correlation serves as an indication of the tension sensor's functionality in measuring inter-filament tensions *in-vivo* and its sensitivity to forces generated by myosin”

By implementing these changes and defining separate terminology for different quantities, we believe we have enhanced the clarity of our manuscript. We genuinely thank the reviewer for their insightful comment again.

2) On a related note, the authors are attempting to use the use the molecular-scale and molecular-specific readout of the FRET-based tension sensor as a stand-in for an intracellular force field (i.e. a macroscopic mechanical variable). This comparison is made in Fig. 3 by comparing the inferred internal stresses and the FRET signal. I do not believe this is a valid/interpretable comparison. This is the equivalent to using the tension on the two arms of a Y-shaped sticks to infer the force on all stick in a giant pile. So, the calculation of the inferred stressed is useful, but the simplest interpretation is that the inferred stresses do not match the tension sensor data b/c one is inferred from the entire force generation of the cell and the other is based on a molecular measurement of one species of molecular crosslinker. While this comparison can be used to emphasize the complexity of the sensor signal, it should not be used to motivate the importance of anisotropy or any particular mechanical variable. It is fine to talk about anisotropy or the presence of F-actin structures. But doing this comparison to motivate the importance of anisotropy when we know there are anisotropic stress fibers in most cells is unnecessary and promotes misuse and misinterpretation of the molecular tension sensor. For this manuscript to be publishable, the strong comparisons of macroscopic stresses and tension sensor output must be removed and replaced with less definitive qualitative conclusions.

Furthermore, I note that reviewer 1 made a similar comment. The fact that the inferred stresses don’t match the tension sensor is also not surprising as the research field has known for decades that cells are not a linearly elastic homogenous solid. Thus, the result that inferred stresses do not match the molecular tension sensors is surprising for a variety of reasons and should not be strongly emphasized.

Thank you for your feedback on Figure 3 and the comparisons made in the manuscript. We appreciate your insights and have taken them into consideration.

The primary goal of Figure 3 is to illustrate the disparities between the **distributions** of molecular tensions and inferred forces from Traction Force Microscopy (TFM). We understand that these quantities are not

directly comparable as the tension sensor does not capture all tension within the cytoskeleton. However, we believe that comparing the distributions is valuable, especially since the sensor localizes to different F-actin structures, which provides insights into the mechanical heterogeneity within the cell. Accordingly, the correlation coefficients between the \bar{F} and FRET, only reflect the relative variations in space, regardless of the magnitudes. To further improve the quality of Figure 3, **we have updated the inferred stress heatmaps to inferred forces**, and the corresponding panels M, N, and O. By focusing on the forces rather than the stresses, we are confident that the comparison is now more justified.

Regarding the last comment, we did not find a related comment from reviewer 1#. We believe reviewer #3 is indeed referring to a comment from reviewer #2 which reads:

“Although the intracellular stress inference is a very interesting approach, the results suggest that it is not relevant as it does not take the mechanical configuration of the cell into account. I suggest pointing this out even more pronounced than already done”.

From reviewer #2's comment, we understand it has been suggested to emphasize more the fact that mechanical configurations are responsible for the observed discrepancies in the tension vs. tractions. We have made the necessary adjustments to the manuscript to address the concerns raised by the reviewers. We appreciate the reviewers' valuable input, which has helped us improve the clarity and interpretation of the data presented in Figure 3.

3) The color bars for the FRET in Fig.3 should be changed. Currently there is a correspondence between 0 Pa Stress and 30% FRET efficiency. However, the 0 pN state of the sensor is 22% FRET efficiency (more on the accuracy of this estimate below). So, if the authors still want to make this comparison, I guess 0 pN and 22% should be matched, at least? But it seems clear that 0 stress and molecular compression should not be matched. I note the vagary in how to set this color table to ensure a viable comparison is generally indicative of the difficulties in comparing molecular tension and macroscopic stress.

We would like to emphasize that **we do not aim to calibrate FRET E with cell-level stresses or forces**. Also, as mentioned in the previous comment, we have transformed the inferred stresses into forces by scaling them with the respective calculation areas. Furthermore, we acknowledge that using a blue-red color bar for the inferred stresses in the previous version could be misleading, as it might imply tension-compression relationships. However, the inferred stresses were always represented as positive values. **Therefore, the color bars have been updated to display a consistent single red color, eliminating any potential confusion.**

We believe these changes have enhanced the visual clarity and accuracy of our presentation, and we thank you for bringing this to our attention.

4) Estimation of G correction factor was done incorrectly due the specifics of the control constructs. Constructs that can be subjected to load cannot be used for this purpose. The premise of the calibration is that the FRET of the two constructs are different and constant. If the constructs can be loaded, the FRET is not constant, and the calibration cannot be performed. The calibration should be re-done with soluble versions of GFP-GSGGS2-RFP and GFP-TRAF-RFP (meaning that there are no flanking domains), as has been done previously. Notably, there are differences between the calibration constructs based on whether they are in stress fibers or the cortex (Sup. Fig. 6 and 7), indicating the variability of the constructs

used in calibration, and clearly demonstrating the need to re-do the calibration. For this paper to be publishable, the unloadable calibration constructs should be produced, used to determine the required calibration factors for determining FRET Efficiency, and then the data reprocessed.

Thank you for raising the concern regarding efficiency control. We acknowledge that the original constructs could potentially introduce variability in the FRET measurements.

To explore this possibility, **we have implemented the soluble calibration constructs**. These new constructs lack any flanking domains, ensuring that they remain free from potential loading effects. Subsequently, we conducted FRET efficiency (FRET E) measurements on the cells expressing the new constructs. We are pleased to report that **the FRET E of these new constructs were found to be insignificantly different from those measured on the previous controls, using the original G values**. This reassuring result cross-validates the two types of control constructs and confirms our original results. As the tension sensor presumably changes its configuration to localize to different actin structures to measure molecular tension, having the actin-bound controls helps us identify the FRET efficiency ranges in each of these distinct F-actin structures. It also helps us rationalize the differences in the intermolecular FRET between these structures. Furthermore, it confirms that the GGSGGS₂ and TRAF linkers are not significantly extended under loading in our experiments.

Therefore, we have kept the main figures, and now have updated Supplementary Figure 6 and the text accordingly. With these revisions and improvements, we are confident in the scientific rigor of our study. We would like to express our gratitude for your valuable feedback, which has significantly contributed to enhancing the quality of our research.

5) I disagree with the authors' interpretation that there is not intermolecular FRET in their system. The TRAF linker is designed as a low FRET control, not a no FRET control. 13% FRET efficiency likely indicates substantial intermolecular FRET. To resolve this issue the authors should co-express GFP and soluble RFP (preferably in a bi-cistronic vector to keep expression levels relatively similar) in their system. The only FRET that can occur in this experiment is bystander FRET. Thus, it will give you an estimate of how a system with very little intermolecular FRET performs. I would anticipate that FRET efficiencies will be less than 5%, a common noise floor in these types of measurements. For this manuscript to be publishable, more definitive evidence that there is a lack of intermolecular FRET in the system must be provided. Without it the contributions of molecular compression versus intermolecular FRET cannot be separated, and the experimental data cannot be interpreted.

We appreciate the reviewer's concern about intermolecular FRET. As the reviewer also mentioned in a previous comment, the dimerized geometry of the tension sensor used in this study is more complex compared to previously developed tension sensors e.g., vinculin. The increased complexity **has enabled us to measure the tension within the F-actin network, however, could lead to higher intermolecular FRET**.

We have shown that the intermolecular FRET at stress fibers is as low as $\approx 6\%$, close to what has been reported [10.1002/cyto.a.23688, doi.org/10.1016/j.cub.2013.04.049]. However, at the cortex, the intermolecular FRET is about $\approx 11\%$. We explain that the **distinct configurations of the tension sensor**

(TS) on cortical actin and stress fibers could be leading to different separations of the fluorophores, presumably between the two dimers, on different F-actin architectures. This spatial difference may result in higher intermolecular FRET at the cortex compared to stress fibers. Therefore, in contrast to previous tension sensors which localize on a single type of protein with high local concentration (e.g., vinculin), our tension sensor may exhibit greater spatial variation in intermolecular FRET. We have now acknowledged this in the discussion.

As the intermolecular FRET level remains below the measured FRET efficiencies, we believe our measurements of molecular compression and its interpretation remain valid, as the **FRET E of (GGSGGS)₂ construct (where the FPs are close to each other),** or the observed **maximum FRET E of the TS construct** is significantly **higher** than the intermolecular FRET. As the tension sensor is bound to the F-actin structure, and RFP and GFPs are not soluble we do not see the usefulness of bystander FRET measurements in this context.

6) The FRET seems substantially lower on the PDMS. A potential reason for this should be given in the text. Also, showing (or at least stating) that CTS sensors report the same FRET Eff on the PA gels used for tractions and the PDMS membranes used for stretch experiments would help alleviate concerns there are technical issues with the experiments. For instance, is it possible that G is different in the two set-ups? If so, G will need to be determined for both.

We thank the reviewer for this comment. We have now added the data regarding the FRET efficiency of the CTS cells on the PDMS substrate, showing the FRET efficiency of CTS cells is consistent with the FRET efficiency on TFM gels (Supplementary Figure 12). Lower FRET efficiencies for cells on PDMS substrates could be explained by higher stiffnesses of the PDMS compared to compliant 2.8 kPa gels used in the traction force measurement. Another possible reason is the fact that PDMS substrates have nanogrooves and hence induce polarization that promotes higher internal forces, and lower FRET efficiencies. We have now added these explanations to the main text. **We confirm that G factors have been separately calculated for each setup.**

MINOR:

1) The authors sometimes refer to the sensor as a filamin sensor and sometimes a modified filamin sensor. This has generated substantial confusion amongst the reviewers. As the sensor is comprised of a relatively small portion of filamin, I might suggest a different name not involving filamin at all or more explicitly pointing out its small fraction. Also, how the sensor is referenced is not the same throughout the manuscript (sometimes it is just called TS). A single term should be used to reference the sensor developed in this work.

Thanks for noting this issue. We have now used 'TS' consistently throughout the manuscript.

2) The author sometimes refer to the sensor as a force sensor. It should not be referred to as a force sensor, as no vectorial information is provided by the sensor. Thus, it should be referred to as a tension sensor.

Fixed.

3) Intermolecular FRET is not referred to as “crosstalk FRET”. In FRET crosstalk is most commonly used to refer to where the donor FP generates signal in acceptor channel and vice versa. This terminology should be removed.

Fixed.

4) In Supp Fig 6, the TRAF domain is drawn incorrectly. It is a structured domain, not a long flexible linker.

Fixed.

REVIEWER COMMENTS

Reviewer #3 (Remarks to the Author):

OVERALL COMMENTS

The authors are striving to improve the manuscript, but there are still some technical issues associated with the use of a small, dimeric tension sensor. The 11% FRET efficiency observed in cortical structures is a strong indication that intermolecular FRET occurs in this compartment. This signal cannot be ignored simply because it is lower FRET than the FRET that occurs in the unloaded TS. Furthermore, the complexity of interpreting the dimeric TS needs to be explained in greater detail in the manuscript. The authors should be readily capable of addressing these concerns, as no new experiments are requested.

MAJOR COMMENTS

1. I disagree with the interpretation of the intermolecular FRET being negligible in this system. The data demonstrates that there is significant intermolecular FRET in the cortical actin compartment, but probably not in stress fiber compartment (11% vs 5% FRET Efficiency, the latter of which is consistent with a frequently documented noise floor). The interpretation of the comparison of the response of cortical and stress fiber compartment to stretch needs to be re-written to discuss the possibility that some of the changes in the FRET signal are likely due to changes in intermolecular FRET between TSMods (likely within a given dimeric TS) as well as intramolecular FRET with each TSMod within a given TSMod happening in one compartment but not the other.
2. The authors are not sufficiently explaining the complexity associated with the interpretation of the dimeric TS. The authors should add a figure, potentially in the supplement, demonstrating how they envision the sensor responding to the normal and shear forces in sensors associated with both parallel and crossed actin filaments. The diagram should account for changes in FRET due to changes in both intermolecular (FRET between two arms of the dimeric sensors) and intramolecular FRET (FRET within a given TSMod) as the filaments are translocated in response to positive and negative forces. One key factor should be acknowledging that close proximity of the filament likely facilitates intermolecular FRET between the two TSMods within a given dimeric TS. Another key point will be distinguishing between molecular compression of the individual TSMods leading to high

FRET (e.g. the FPs with a given TSMOD are driven closer together) as well as the ability of the proximity of the filaments to induce intermolecular FRET between TSMODs in a given dimeric TS. These separate processes are currently both referred to as compression in the text. They should also likely be referred to different processes (see minor comment 4 below).

3. As intermolecular FRET between the TSMODs in the dimeric TS could be present even if the TS is loaded, the presence of substantial intermolecular FRET invalidates the previously established calibration for the TSMOD. A sentence stating this should be added to the text.

4. There is no indication of the baseline FRET before stretching in many of experiments. The determination of net compression or tension is a powerful approach and is key to the main observations of the paper. But, knowing if the inhibitors completely unloaded the TS is also of value. The baseline FRET before stretching for all experimental conditions should be added to the manuscript as a supplemental figure.

MINOR COMMENTS

1. Text on line 70 needs refinement. "Under tensile loads of up to 20 pN". The working range of the genetically encoded tension sensors is typically 1-6 pN, not 1-20 pN. That is the range given in the cited paper as well.

2. The sentences beginning on line 123 needs refining. The ability of the molecular tension sensor to report loads within the F-actin cytoskeleton is not dependent on a relationship to magnitude of the traction vectors. The sensor could also be functional in a suspended cell. This section simply probes the correlation between the tension sensor signal and traction force. This assumes the proper function of the tension sensor and should be stated as such.

3. Line 148: The FRET between a bound TS and unbound TS is a version of the bystander FRET. This only occurs at the highest of expression levels and is dependent on the concentration of the TS. The authors present evidence that the FRET efficiency is not dependent on concentration/expression levels. Thus, a much more likely scenario is that there is significant intermolecular FRET between the two TSMODs in a given dimeric TS. This signal is much more likely to be independent of concentration, but dependent on the local arrangement of F-actin filaments, which is what the authors observe. Also, it seems likely that the "arms" of the TS might be able to bind in closer proximity on crossed F-actin filaments, potentially explaining the intermolecular FRET in cortex, but not SFs. These points

should be added to the text. Thus, the sentence should be edited to discuss intermolecular FRET within in a single dimer and not bystander FRET between one bound and one unbound TS.

4. Line 177-181: Molecular and mesoscale mechanical variables are conflated in this section. I agree with the statement that FRET levels above CTS levels could be due to forcing the FPs within an individual TSMOD closer than the rest length as well as the increase of intermolecular FRET due to FRET between two TSMODs in response to the reduced spacing of two F-actin filaments due to compression. However, these are distinct processes. One way to acknowledge this is to consider molecular tension as varying from negative to positive and referring to FRET signal changes due to changes in intramolecular FRET within an individual TS in this manner. Then “compression” could refer to a mesoscale effect of actin filament separation distance increase the intermolecular FRET between the two TSMODs with a single TS.

5. The sentence starting in line 190 and continuing to 191 describing the lack of a unique relationship between FRET signals and traction should be edited to include the possibility that one compartment may have a contribution from intermolecular FRET and one may not.

6. Line 251: The phrase “does not bear a net compression or tension” is potentially confusing as it could be readily confused with not bearing tension or compression. Saying “do not exhibit changes in net tension” would alleviate this concern.

OVERALL COMMENTS

The authors are striving to improve the manuscript, but there are still some technical issues associated with the use of a small, dimeric tension sensor. The 11% FRET efficiency observed in cortical structures is a strong indication that intermolecular FRET occurs in this compartment. This signal cannot be ignored simply because it is lower FRET than the FRET that occurs in the unloaded TS. Furthermore, the complexity of interpreting the dimeric TS needs to be explained in greater detail in the manuscript. The authors should be readily capable of addressing these concerns, as no new experiments are requested.

We thank the reviewer and appreciate their insightful comments and suggestions. In the revised version, we have provided a more detailed explanation to interpreting the tension sensor results. We have tried our best to address all the comments and strongly believe their suggestions have significantly improved the quality of the manuscript since the initial submission.

MAJOR COMMENTS

1. I disagree with the interpretation of the intermolecular FRET being negligible in this system. The data demonstrates that there is significant intermolecular FRET in the cortical actin compartment, but probably not in stress fiber compartment (11% vs 5% FRET Efficiency, the latter of which is consistent with a frequently documented noise floor). The interpretation of the comparison of the response of cortical and stress fiber compartment to stretch needs to be re-written to discuss the possibility that some of the changes in the FRET signal are likely due to changes in intermolecular FRET between TSMods (likely within a given dimeric TS) as well as intramolecular FRET with each TSMoD within a given TSMoD happening in one compartment but not the other.

We thank the reviewer for this insightful comment. We have elaborated more on the possibilities of changes in intermolecular FRET during stretching at the stress fibers or cortical actin. We would like to also mention that as intermolecular FRET is a function of sensor expression, and as we have not seen significant change in FRET efficiency with different expression levels (Supplementary Figure 9), we suggest the inter/intramolecular FRET is not playing a major role in our measurements. Therefore, now we have revised the related section as follows:

“For example, it may be possible that the monomers within a single TS construct bind in closer proximity at the cortex compared to stress fibers, potentially resulting in increased intermolecular FRET (Supplementary Figure 16)⁶⁰. Nevertheless, as intermolecular FRET is a function of sensor expression levels^{33,66}, showing that base line FRET efficiencies are not significantly regulated by the TS expression levels (Supplementary Figure 9), suggests intermolecular FRET is not obscuring the FRET signal. In any event, we have considered potential effects that the inter/intramolecular FRET may have on our FRET E measurements.”

We have also revised the last section considering the inter/intramolecular at different compartments as follows:

“The decrease in FRET E can also occur due to a decrease in inter/intramolecular FRET between dimers of TS. In stress fiber sliding under shear forces TS molecules move away from each and likely change their

configuration (Supplementary Figure 16B). On the cortical actin, normal and shear forces likely increase the distance between dimeric TS sensors, as well as increasing tension on the individual TS (Supplementary Figure 16A). Conversely, in orthogonal cells, Δ FRET E of the SFs increase, while decreasing for the cortical actin (Fig. 5C). The increase in FRET E of the SFs is a result of their relaxation, as well as a potential increase in the intermolecular FRET of the dimeric tension sensor constructs by relaxation of the dimers getting closer together (Supplementary Figure 16 B). Although there is a larger level of inter/intramolecular FRET in cortical actin, we observed a reduction in FRET efficiency. This is potentially a consequence of increased tension and a decrease in inter/intramolecular FRET in the stretched cortex as the dimers of TS move away under stretch (Supplementary Figure 16A). Again, the different change in Δ FRET E is indicative of mechanical anisotropy of the cell and differential load bearing in SFs and the cortical actin.”

“Higher Δ FRET E of SFs is suggestive of higher accumulation of molecular tensions in SFs compared to the cortical actin. This can also be reflective of how intermolecular FRET changes in these compartments (Supplementary Figure 16). The slopes were significantly different ($P < 0.0001$), suggesting an anisotropic mechanical response, and differential load bearing between SFs and the cortical actin.

To evaluate the influence of myosin-generated forces and the intermolecular FRET in our measurements, we conducted additional analyses using ML7 and Y27632 drugs while retaining the presence of the cortex and SFs. Upon inhibition of myosin II activity using 67 μ M ML7 (Fig. 5E) or 10 μ M Y27632 (Fig. 5F), SFs were still present. Again, a strong correlation was observed for ML7 ($m \approx 0.64$ and $m \approx 0.62$ in parallel or orthogonal directions) or Y27632 ($m \approx 0.74$ and $m \approx 0.75$ in parallel or orthogonal directions) -treated cells. However, the difference in the slopes was nonsignificant in two directions. Assuming intermolecular FRET is consistent between treated and non-treated SFs and cortical networks, our results indicate that in the absence of myosin activity, SFs accumulate less molecular tension compared to cortical actin. These results indicated that SFs and cortical regions exhibit differential load bearing depending on the direction of strain (Fig. 5G).”

2. The authors are not sufficiently explaining the complexity associated with the interpretation of the dimeric TS. The authors should add a figure, potentially in the supplement, demonstrating how they envision the sensor responding the normal and shear forces in sensors associated with both parallel and crossed actin filaments. The diagram should account for changes in FRET due changes in both intermolecular (FRET between two arms of the dimeric sensors) and intramolecular FRET (FRET within a given TSMOD) as the filaments are translocated in response to positive and negative forces. One key factor should be acknowledging that close proximity of the filament likely facilitates intermolecular FRET between the two TSMODs within a given dimeric TS. Another key point will be distinguishing between molecular compression of the individual TSMODs leading to high FRET (e.g. the FPs with a given TSMOD are driven closer together) as well as the ability of the proximity of the filaments to induce intermolecular FRET between TSMODs in a given dimeric TS. These separate processes are currently both referred to as compression in the text. They should also likely be referred to different processes (see minor comment 4 below).

We thank the reviewer for their valuable insights. We have now included a schematic of the TS on the cortex and stress fibers, under normal and shear forces in Supplementary Figure 16. This diagram illustrates higher inter/intramolecular FRET on the cortical actin compared to the stress fibers, due to the crosslinked nature

of the cortex and their closer proximity. We also show that on the stress fibers, normal forces can lead to higher intermolecular FRET because of the proximity of the TS, as well as higher intermolecular and compression FRET. Under shear forces, by relative sliding of SF filaments, dimerized TS is likely to elongate leading to a decrease in inter/intramolecular FRET as well as an increase in tension. On the cortex, with normal or shear forces, TS undergo tension, which also decreases their inter/intramolecular FRET.

3. As intermolecular FRET between the TSMods in the dimeric TS could be present even if the TS is loaded, the presence of substantial intermolecular FRET invalidates the previously established calibration for the TSMod. A sentence stating this should be added to the text.

We thank the author for their insightful comment. We have added the following sentence to address this issue:

“It should be noted that the presence of intermolecular FRET between the dimeric tension sensor (TS), could potentially render the previous calibrations invalid for this investigation.”

4. There is no indication of the baseline FRET before stretching in many of experiments. The determination of net compression or tension is a powerful approach and is key to the main observations of the paper. But, knowing if the inhibitors completely unloaded the TS is also of value. The baseline FRET before stretching for all experimental conditions should be added to the manuscript as a supplemental figure.

We thank the reviewer for this comment. We have now updated Supplementary Figure 12 to include the baseline FRET values for all the experimental conditions. The FRET values in the relax state indicate that inhibitors do not completely unload the TS constructs, but due to stretching, the TS experiences more tension or relaxation (from a tense state). We also notice the differences are more significant by calculation of relative FRET changes. We explain the reason for the difference in significance due to the inter cellular variability in the FRET efficiencies. When we calculate relative changes, differences in the initial FRET values lead to significant changes, whereas the absolute differences might not show the same level of significance because they are not adjusted for the initial values.

We have now added: “Baseline FRET E values indicated that the TS is not completely unloaded under different inhibitory conditions (Supplementary Figure 12). The TS in cells under Inhibitory drugs exhibits a baseline tension, which undergoes relaxation or increased tension depending on the direction or magnitude of applied strain. We also notice the differences in FRET are more significant by calculation of relative FRET changes. We explain the reason for the difference in significance due to the intercellular variability in the FRET efficiencies as they are not adjusted for the relaxed FRET values.”

MINOR

COMMENTS

1. Text on line 70 needs refinement. “Under tensile loads of up to 20 pN”. The working range of the genetically encoded tension sensors is typically 1-6 pN, not 1-20 pN. That is the range given in the cited paper as well.

Fixed.

2. The sentences beginning on line 123 needs refining. The ability or the molecular tension sensor to report loads within the F-actin cytoskeleton is not dependent on a relationship to magnitude of the traction vectors. The sensor could also be functional in a suspended cell. This section simply probes the correlation between the tension sensor signal and traction force. This assumes the proper function of the tension sensor and should be stated as such.

We have revised the sentence which now reads:

“To assess the sensitivity of the molecular tensions within the F-actin cytoskeleton to traction forces, we compared the changes in mean FRET efficiency within cells ($\langle \text{FRET } E \rangle$, Material and Methods) to the magnitude of traction vectors...”

3. Line 148: The FRET between a bound TS and unbound TS is a version of the bystander FRET. This only occurs at the highest of expression levels and is dependent on the concentration of the TS. The authors present evidence that the FRET efficiency is not dependent on concentration/expression levels. Thus, a much more likely scenario is that there is significant intermolecular FRET between the two TSMods in a given dimeric TS. This signal is much more likely to be independent of concentration, but dependent on the local arrangement of F-actin filaments, which is what the authors observe. Also, it seems likely that the “arms” of the TS might be able to bind in closer proximity on crossed F-actin filaments, potentially explaining the intermolecular FRET in cortex, but not SFs. These points should be added to the text. Thus, the sentence should be edited to discuss intermolecular FRET within a single dimer and not bystander FRET between one bound and one unbound TS.

We thank the reviewer for this point. The revised sentence now refers to the intermolecular FRET within a dimeric tension sensor:

“For example, it may be possible that compared to stress fibers, the dimers within a single TS construct bind in closer proximity at the cortex, potentially resulting in increased intermolecular FRET.”

4. Line 177-181: Molecular and mesoscale mechanical variables are conflated in this section. I agree with the statement that FRET levels above CTS levels could be due to forcing the FPs within an individual TSMoD closer than the rest length as well as the increase of intermolecular FRET due to FRET between two TSMods in response to the reduced spacing of two F-actin filaments due to compression. However, these are distinct processes. One way to acknowledge this is to consider molecular tension as varying from negative to positive and referring to FRET signal changes due to changes in intramolecular FRET within an individual TS in this manner. Then “compression” could refer to a mesoscale effect of actin filament separation distance increase the intermolecular FRET between the two TSMods with a single TS.

We thank the reviewer for the careful read. We have now revised the section which reads:

“The FRET efficiency of the sensor spans a range from molecular tension (FRET $E \approx 15\%$) to molecular compression (FRET $E \approx 33\%$). The transition from tension to compression occurs at the rest length of the F40 linker with approximately $\approx 22 \pm 2\%$ corresponding to the FRET E of the load-free CTS, which appears to be comparable to the FRET E of the TS ($\approx 24\% \pm 2\%$) at $\langle |\vec{T}| \rangle \approx 0$. Thus, depending on the value of FRET E , an increase in FRET E can result from two distinct factors: the fluorophores moving closer together than the rest length of the F40 linker, and an elevated level of intermolecular FRET caused

by FRET interactions between dimers within a single TS. In this case, the actin filaments may be undergoing compression as they move closer together.”

5. The sentence starting in line 190 and continuing to 191 describing the lack of a unique relationship between FRET signals and traction should be edited to include the possibility that one compartment may have a contribution from intermolecular FRET and one may not.

It now reads:

“This could be due to differences in intermolecular FRET within distinct actin architectures induced by various drugs, F-actin content, cell-ECM coupling^{66,67}, or cell stiffness⁶⁸”

6. Line 251: The phrase “does not bear a net compression or tension” is potentially confusing as it could be readily confused with not bearing tension or compression. Saying “do not exhibit changes in net tension” would alleviate this concern.

Fixed.

REVIEWERS' COMMENTS

Reviewer #3 (Remarks to the Author):

I commend the authors for their changes to the paper. It could be published in this form, but I might suggest a small change to the paper to eliminate the confusion that the authors and I have been having. To their credit, I was able to figure out the issue due to the addition of Supp Fig 16.

There are very few dimeric FRET-based tension sensors containing two tension sensitive modules. Many of the issues we have been discussing during the review process come down to how myself and the reviews have been using the term “intermolecular FRET”. In monomeric tension sensors with one module, there are only two types of FRET: FRET within the sensor and FRET between two sensors. These are termed intramolecular and intermolecular FRET. Commonly, this type of intermolecular FRET is referred to as “bystander FRET”, and does indeed involve a concentration dependence if the protein of interest is not subject to clustering or multimerization.

This scenario become more complex in a dimeric sensor contain two tension sensitive modules, as has been developed in this manuscript. Now there are three types of FRET: FRET within each monomeric “arm of the sensor”, FRET between the two monomeric “arms” of a single dimeric sensor, and FRET between two separate dimeric sensors. Basically, I was viewing the monomer as a single “molecule”. So, I was identifying the first class as intramolecular FRET and then grouping the second and third classes as intermolecular FRET. The authors were viewing the dimer as a single molecule, and thus grouping the first two types of FRET as intramolecular, and then the third as intermolecular FRET.

To prevent confusion in the field, I suggest the authors create a new terminology to describe the types of FRET possible in their construct with three distinct terms. I might suggest naming the three types of FRET described above as intramonomer FRET, intradimer FRET, and then interdimer FRET. Intramonomer FRET is the type of FRET that the tension sensors were initially designed to study and the calibrations to convert FRET Eff into tensions work

for. Intradimeric FRET is likely concentration-independent (as it is occurring within a single sensor), but can be dependent on mechanical loads. This is now well-addressed in the manuscript. Interdimeric FRET is analogous to standard bystander FRET, if there is no clustering of sensors. As this is for the most part a synthetic sensor, it is reasonable to assume that clustering may be ignored for this sensor. Thus, there is likely to be a concentration dependence to intradimeric FRET. The presence of either intradimeric or interdimeric FRET in a system invalidates the calibrations for turning FRET efficiency to molecular tensions.

So with this terminology in place we can then discuss the data in the paper. As the authors suggest, the lack of a concentration dependence suggests there is little intradimeric, bystander FRET. However the 11% FRET efficiency suggests there is substantial intramonomer FRET in some sub-cellular compartments. This is now addressed properly throughout the manuscript.

So in summary, I believe this is an important terminology to develop for the field. If it is not established now, I fear that the miscommunication happening in this review will then spread through the field. So, I would request that the authors create three terms for the type of FRET in their sensor (they can use the terminology or develop another, as they see fit) and then apply them to Fig S16 and the rest of the manuscript. I feel that it is important to state that I am not critiquing their methods, data, or conclusions. In fact, I commend them for the receptiveness and rigorousness through which they have responded to reviewer concerns.

Reviewer #3 (Remarks to the Author):

I commend the authors for their changes to the paper. It could be published in this form, but I might suggest a small change to the paper to eliminate the confusion that the authors and I have been having. To their credit, I was able to figure out the issue due to the addition of Supp Fig 16.

There are very few dimeric FRET-based tension sensors containing two tension sensitive modules. Many of the issues we have been discussing during the review process come down to how myself and the reviews have been using the term “intermolecular FRET”. In monomeric tension sensors with one module, there are only two types of FRET: FRET within the sensor and FRET between two sensors. These are termed intramolecular and intermolecular FRET. Commonly, this type of intermolecular FRET is referred to as “bystander FRET”, and does indeed involve a concentration dependence if the protein of interest is not subject to clustering or multimerization.

This scenario become more complex in a dimeric sensor contain two tension sensitive modules, as has been developed in this manuscript. Now there are three types of FRET: FRET within each monomeric “arm of the sensor”, FRET between the two monomeric "arms" of a single dimeric sensor, and FRET between two separate dimeric sensors. Basically, I was viewing the monomer as a single “molecule”. So, I was identifying the first class as intramolecular FRET and then grouping the second and third classes as intermolecular FRET. The authors were viewing the dimer as a single molecule, and thus grouping the first two types of FRET as intramolecular, and then the third as intermolecular FRET.

To prevent confusion in the field, I suggest the authors create a new terminology to describe the types of FRET possible in their construct with three distinct terms. I might suggest naming the three types of FRET described above as intramonomer FRET, intradimer FRET, and then interdimer FRET. Intramonomer FRET is the type of FRET that the tension sensors were initially designed to study and the calibrations to convert FRET Eff into tensions work for. Intradimeric FRET is likely concentration-independent (as it is occurring within a single sensor), but can be dependent on mechanical loads. This is now well-addressed in the manuscript. Interdimeric FRET is analogous to standard bystander FRET, if there is no clustering of sensors. As this is for the most part a synthetic sensor, it is reasonable to assume that clustering may be ignored for this sensor. Thus, there is likely to be a concentration dependence to intradimeric FRET. The presence of either intradimeric or interdimeric FRET in a system invalidates the calibrations for turning FRET efficiency to molecular tensions.

So with this terminology in place we can then discuss the data in the paper. As the authors suggest, the lack of a concentration dependence suggests there is little intradimeric, bystander FRET. However the 11% FRET efficiency suggests there is substantial intramonomer FRET in some sub-cellular compartments. This is now addressed properly throughout the manuscript.

So in summary, I believe this is an important terminology to develop for the field. If it is not established now, I fear that the miscommunication happening in this review will then spread through the field. So, I would request that the authors create three terms for the type of FRET in their sensor (they can use the terminology or develop another, as they see fit) and then apply them to Fig S16 and the rest of the manuscript. I feel that it is important to state that I am not critiquing their methods, data, or conclusions. In fact, I commend them for the receptiveness and rigorousness through which they have responded to reviewer concerns.

We are grateful for your positive feedback. We appreciate your efforts to clarify the complexities of our dimeric FRET-based tension sensor and your suggestion to clarify the terminology. Based on your suggestions, we have adopted the terms "intramonomeric FRET" to describe FRET occurring within each monomer, "intradimeric FRET" to represent FRET between the two monomeric arms of a single dimeric sensor, and "interdimeric FRET" to denote FRET between two separate dimeric sensors. We believe these terms accurately reflect the distinct FRET interactions within our construct and clarify the terminology. We have accordingly revised the manuscript, and incorporated these terms consistently throughout the text, including in Supplementary Figure 16, and Supplementary Figure 7.